Resource

# iMPAQT reveals that adequate mitohormesis from TFAM overexpression leads to life extension in mice

Ko Igami[1,2,3] , Hiroki Kittaka[1,2], Mikako Yagi[3,4] , Kazuhito Gotoh[3,5], Yuichi Matsushima[3,6] , Tomomi Ide[7], Masataka Ikeda[7] , Saori Ueda[3], Shin-Ichiro Nitta[1,2], Manami Hayakawa[2], Keiichi I. Nakayama[8,9] , Masaki Matsumoto[10] , Dongchon Kang[3,11], Takeshi Uchiumi[3,4]

**Mitochondrial transcription factor A, TFAM, is essential for mitochondrial function. We examined the effects of overexpressing the *TFAM* gene in mice. Two types of transgenic mice were created: *TFAM* heterozygous (*TFAM* Tg) and homozygous (*TFAM* Tg/Tg) mice. *TFAM* Tg/Tg mice were smaller and leaner notably with longer lifespans. In skeletal muscle, TFAM overexpression changed gene and protein expression in mitochondrial respiratory chain complexes, with down-regulation in complexes 1, 3, and 4 and up-regulation in complexes 2 and 5. The iMPAQT analysis combined with metabolomics was able to clearly separate the metabolomic features of the three types of mice, with increased degradation of fatty acids and branched-chain amino acids and decreased glycolysis in homozygotes. Consistent with these observations, comprehensive gene expression analysis revealed signs of mitochondrial stress, with elevation of genes associated with the integrated and mitochondrial stress responses, including Atf4, Fgf21, and Gdf15. These found that mitohormesis develops and metabolic shifts in skeletal muscle occur as an adaptive strategy.**

## Introduction

Mitochondria regulate a multitude of different signaling and metabolic pathways, such as energy production, fatty acid and amino acid oxidation, iron metabolism, and apoptosis (Schmidt et al, 2010). However, mitochondria also generate reactive oxygen species as byproducts of oxidative phosphorylation. Thus, mitochondrial DNA (mtDNA) is more exposed to oxidative stress than nuclear DNA (Kang & Hamasaki, 2005a). Mitochondrial transcription factor A (TFAM) was cloned as an mtDNA transcription factor by Fisher and Clayton (Fisher & Clayton, 1988). TFAM binds mtDNA and forms a nucleoid-like structure, protecting mtDNA from DNA-damaging agents such as reactive oxygen species (Alam et al, 2003; Kanki et al, 2004; Kang & Hamasaki, 2005b). Therefore, TFAM plays an essential role in mtDNA stability by regulating mtDNA replication, transcription, and packaging (Gaspari et al, 2004; Kang et al, 2007; Campbell et al, 2012).

Mice overexpressing human TFAM (hTFAM) under a modified chicken β-actin promoter show amelioration of the decreases in mtDNA copy number and mitochondrial complex enzyme activities in post-myocardial infarction hearts (Ikeuchi et al, 2005), and show dramatically improved symptoms in mouse models of transient forebrain ischemia (Hokari et al, 2010) and amyotrophic lateral sclerosis (Morimoto et al, 2012). Fujii et al showed that hTFAM homozygous transgenic (Tg/Tg) mice had highly activated brown adipocytes and an increased expression of oxidative phosphorylation, leading to resistance to obesity (Fujii et al, 2022).

Various biological phenotypes have been observed in mice overexpressing TFAM, but the underlying mechanisms in the tissues that directly express the protein are unknown. In addition to the well-studied phenotype of TFAM heterozygous overexpression, we also aimed to observe TFAM homozygous overexpression. In this study, we analyze skeletal muscles in which hTFAM is directly expressed because of its promoter characteristics. A comprehensive omics approach is very helpful in understanding the processes occurring. Because mitochondria are particularly involved in biological functions, metabolic shifts are expected. This study was based on an omics analysis of proteins and metabolites. For the proteomics, we implemented a mass spectrometry method called in vitro proteome-assisted multiple reaction monitoring for Protein Absolute QuanTification (iMPAQT), which enables precise and absolute quantification (Matsumoto et al, 2017). Before this study, we

[1]LSI Medience Corporation, Tokyo, Japan    [2]Kyushu Pro Search Limited Liability Partnership, Fukuoka, Japan    [3]Department of Clinical Chemistry and Laboratory Medicine, Kyushu University Graduate School of Medical Sciences, Fukuoka, Japan    [4]Clinical Chemistry, Division of Biochemical Science and Technology, Department of Health Sciences, Faculty of Medical Sciences, Kyushu University, Fukuoka, Japan    [5]Department of Laboratory Medicine, Tokai University School of Medicine, Kanagawa, Japan    [6]Department of Biological Sciences, Graduate School of Science, Osaka University, Toyonaka, Japan    [7]Department of Cardiovascular Medicine, Graduate School of Medical Sciences, Kyushu University, Fukuoka, Japan    [8]Department of Molecular and Cellular Biology, Medical Institute of Bioregulation, Kyushu University, Fukuoka, Japan    [9]Anticancer Strategies Laboratory, TMDU Advanced Research Institute, Tokyo Medical and Dental University, Tokyo, Japan    [10]Department of Omics and Systems Biology, Graduate School of Medical and Dental Sciences, Niigata University, Niigata, Japan    [11]Kashiigaoka Rehabilitation Hospital, Fukuoka, Japan

Correspondence: igami.ko@mu.medience.co.jp; uchiumi.takeshi.008@m.kyushu-u.ac.jp

developed a novel iMPAQT method that can measure over 1,100 mouse metabolic proteins. In the iMPAQT assay, quantification is performed using standard peptides with stable isotopes incorporated into lysine and arginine so that results are derived in terms of the number of molecules (fmol/μg). This allows for quantitative determination of very fine details of the enzymes in metabolic pathways. Taking advantage of this very powerful method, we analyzed skeletal muscle tissues of WT, heterozygous, and homozygous mice, and found that adequate mitohormesis resulting from TFAM overexpression leads to healthy life extension. We discuss the metabolic and proteomic shifts in relation to the unique phenotypes of the TFAM-overexpressing mice.

# Results

### Phenotype of TFAM-overexpressing mice and the effects on mitochondrial respiratory chain complexes in skeletal muscle

For this study, we prepared two types of transgenic mice, human TFAM heterozygous (TFAM Tg) and homozygous (TFAM Tg/Tg) mice, as well as the WT. The TFAM Tg/Tg mice were leaner and smaller in stature than the other mice (Fig 1A and B), and there were no apparent differences between WT and TFAM Tg mice. The gastrocnemius muscle of TFAM Tg/Tg mice was smaller than that of WT and TFAM Tg mice (Fig 1C). Dissection revealed less fat, and they also had low levels of free fatty acids in their blood (Fig S1A and B), which may be one reason for their smaller body size. Also shown in Fig 1D is the overall survival of these three types of mice, with TFAM Tg/Tg mice tending to have a longer lifespan than WT mice (median survival times: WT, 872 d; Tg, 937 d; Tg/Tg, 1,007 d). On the contrary, the TFAM transgene was located in intron 1 of the Irak3 (IRAK-M) gene on chromosome 10 (Ylikallio et al, 2010). Therefore, we confirmed the protein level of IRAK-M expression in gastrocnemius muscle by Western blotting and found no changes in both Tg and Tg/Tg (Fig 1E). Using RNA extracted from the gastrocnemius muscle of these mice, we confirmed the mRNA levels of the incorporated hTFAM gene, which were more than twice as high in TFAM Tg/Tg mice than in TFAM Tg mice (Fig 1F).

We investigated the mRNA levels of each gene encoded in the mtDNA (Fig 1G) and the protein expression of the respiratory chain complex (Figs 1H and I and S2) because of TFAM overexpression in skeletal muscle. The overexpression of hTFAM reduced the expression of genes in the respiratory chain complexes (other than ribosomal RNA) relative to WT (Fig 1G). Furthermore, we performed quantitative protein analysis using the iMPAQT method. Proteins involved in respiratory chain complexes 1, 3, and 4 were uniformly decreased in TFAM Tg and Tg/Tg (Figs 1H, J, and K and S2). For complex 2, they were increased in Tg/Tg (Figs 1I and S2). For complex 5 (F-type ATPase), the expression of each subunit was increased, in contrast to the mRNA levels (Figs 1L and S2). TFAM expression had no effect on the V-type ATPase (Fig S2). The overexpression of TFAM had a repressive effect on the expression of genes and proteins encoded by the mtDNA. With respect to the respiratory chain complexes, there was a tendency to increase protein levels and their subunits derived from genes encoded in the nucleus.

### Comprehensive gene expression analysis revealing mitochondrial stress in TFAM-overexpressing skeletal muscle

The overexpression of TFAM results in significant changes in mtDNA-encoded gene expression and proteins in the respiratory chain complex, suggesting that there are several stresses on the mitochondria, which may result in changes in various transcriptional cascades. A comprehensive gene expression analysis by DNA microarray was performed using RNA extracted from the gastrocnemius muscle (Supplemental Data 1). Gene set enrichment analysis (GSEA) was then performed using the Gene Transcription Regulation Database (GTRD, gtrd.biouml.org) to derive potential regulatory targets for each differentially expressed transcription factor. The top five categories of enrichment ± scores extracted in the comparison between WT and Tg/Tg are shown (Fig 2A).

The CCAAT/enhancer-binding protein (C/EBP) family is a group of leucine zipper transcription factors consisting of C/EBP-$α$, $β$, $δ$, $γ$, and $ε$. C/EBP-$γ$ is essential for the induction of the integrated stress response (ISR) (Renfro et al, 2022). The specificity in the transcriptional response to ISR is provided by the heterodimeric partner of activating transcription factor 4 (ATF4) (Huggins et al, 2016; Pakos-Zebrucka et al, 2016). In the GSEA results, the group of genes targeted by C/EBP-$γ$ ranked the highest positively (up-regulated in Tg/Tg), which also included FGF21 (Fig 2B). DDIT3/CHOP mediate the mitochondrial stress response (Zhao et al, 2002). A mechanism has also been proposed in which DDIT3/CHOP fine-tunes mitochondrial ISR in mammals (Kaspar et al, 2021). The genes targeted by DDIT3 in our results were also significantly enriched (Fig 2C). qRT–PCR analysis of ISR, including the mitochondrial stress activation genes Atf4, Fgf21, Gdf15, Sestrin2, and Ddit3/Chop, was performed using RNA extracted from the gastrocnemius muscle. A predominant increase in these genes was observed in Tg/Tg (Fig 2D–H).

SRY-box transcription factor 6 (SOX6) is a multifaceted transcription factor involved in terminal differentiation. In skeletal muscle, SOX6 has been suggested to be an important factor in muscle development (An et al, 2011; Quiat et al, 2011), playing a role in regulating muscle differentiation and promoting the development of satellite cell–derived myotubes into inherent muscle fiber types (Zhang et al, 2022). The GSEA results suggested that the gene cluster targeted by SOX6 is down-regulated in the skeletal muscle of Tg/Tg mice (Fig 2I). Therefore, in the skeletal muscle of Tg/Tg mice, mitochondrial stress may be the starting point for the activation of muscle atrophy–inducing transcription factors.

### Comparative analysis of skeletal muscle from WT and TFAM-overexpressing mice by iMPAQT and metabolomics

Next, to unravel the molecular mechanisms directly involved in the mouse phenotypes, quantitative analysis of metabolic proteins by the iMPAQT method and metabolomics was performed on gastrocnemius muscle from the three genotypes of mice. The iMPAQT method was able to detect 247 proteins involved in metabolism from proteins extracted from the gastrocnemius muscle of five animals of each type (Supplemental Data 1). The results, shown in Fig 3A by principal component analysis (PCA), show that the three types of mice can be segregated by group. A comprehensive

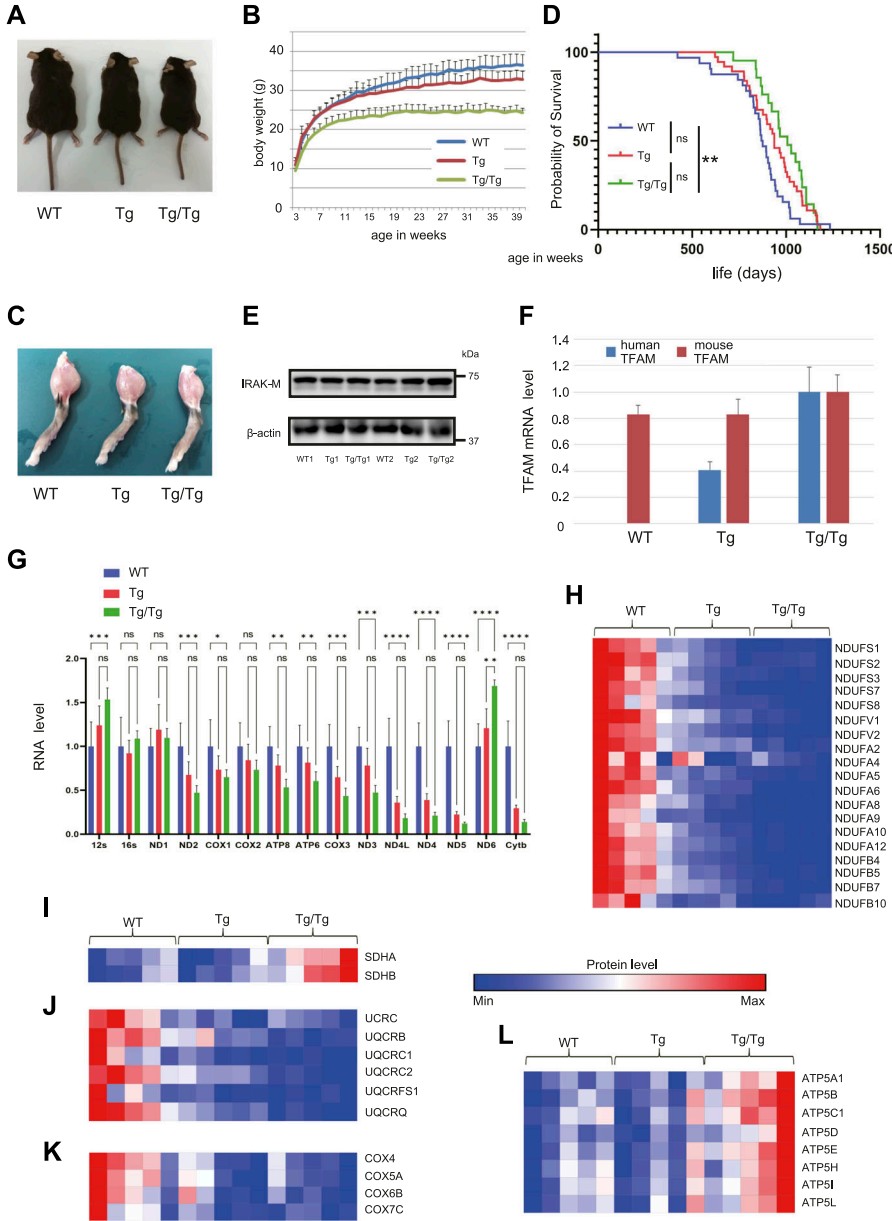

**Figure 1. TFAM overexpression phenotype and its effect on respiratory chain complexes.**
**(A)** Representative examples of WT and TFAM Tg and Tg/Tg mice. **(B)** Graph showing the weekly age and body weight of each mouse (WT, n = 6; Tg, n = 7; Tg/Tg, n = 7). Error bars are presented as the mean + SD of mice in each group. **(C)** Examples of the gastrocnemius muscle of WT and TFAM Tg and Tg/Tg mice used in this study. **(D)** Survival proportions (lifespans) of the three types of mice are shown (WT, n = 32; Tg, n = 37; and Tg/Tg, n = 21). The log-rank Mantel–Cox test was used to compare survival curves. **P < 0.01. **(E)** Western blot revealed no change in IRAK-M expression in three types of mouse skeletal muscle. **(F, G)** Using RNA extracted from the gastrocnemius muscle from the three types of mice, qRT–PCR was performed on (F) the human *Tfam* and mouse *Tfam* genes and (G) mtDNA-encoded genes (n = 4 mice for each group). Error bars are presented as the mean + SD of mice in each group. *P < 0.05, **P < 0.01, ***P < 0.001, ****P < 0.0001 versus Tg/Tg mice. **(G)** uses two-way ANOVA with Tukey's multiple comparisons test. Quantification of mitochondrial respiratory chain complex proteins extracted from the three types of gastrocnemius muscle by the iMPAQT method. **(H, I, J, K, L)** Shown in the heat map are (H) complex 1, (I) complex 2, (J) complex 3, (K) complex 4, and (L) complex 5 F-type ATPase (n = 5 mice for each group).

analysis of metabolites from 10 mice of each type was also performed by GC-MS and LC-MS, and 327 metabolites were detected (Supplemental Data 1). Fig 3B shows the PCA results, with some individuals slightly different from the other types in Tg/Tg, but the results were not separable among the groups. Fig 3C shows a volcano plot of the iMPAQT results. Differentially expressed proteins were selected using the threshold values of |fold change| > 1.5 and t test P-value < 0.05. As mentioned earlier, proteins involved in oxidative phosphorylation were commonly decreased in the comparisons between WT and Tg and between WT and Tg/Tg, but proteins involved in fatty acid degradation and branched-chain amino acid (BCAA: valine, leucine, and isoleucine) degradation were increased only in the comparison between WT and Tg/Tg. The list of proteins other than the respiratory chain complex and metabolites

that had P-values < 0.05 in the comparison between WT and Tg/Tg is shown in the heat map (Figs 3D and S3).

Furthermore, we performed an integrative analysis in MetaboAnalyst using the results of the proteins and metabolites selected here. The results are shown in Fig 3E. The top pathways selected here were those involved in the final stages of glycolysis, in addition to fatty acid degradation and BCAA degradation.

## Enhanced fatty acid degradation pathway in the skeletal muscle of TFAM-overexpressing mice

Among the pathways highly ranked by integrated proteomics and metabolomics, we focused on fatty acid degradation. Fig 4A shows

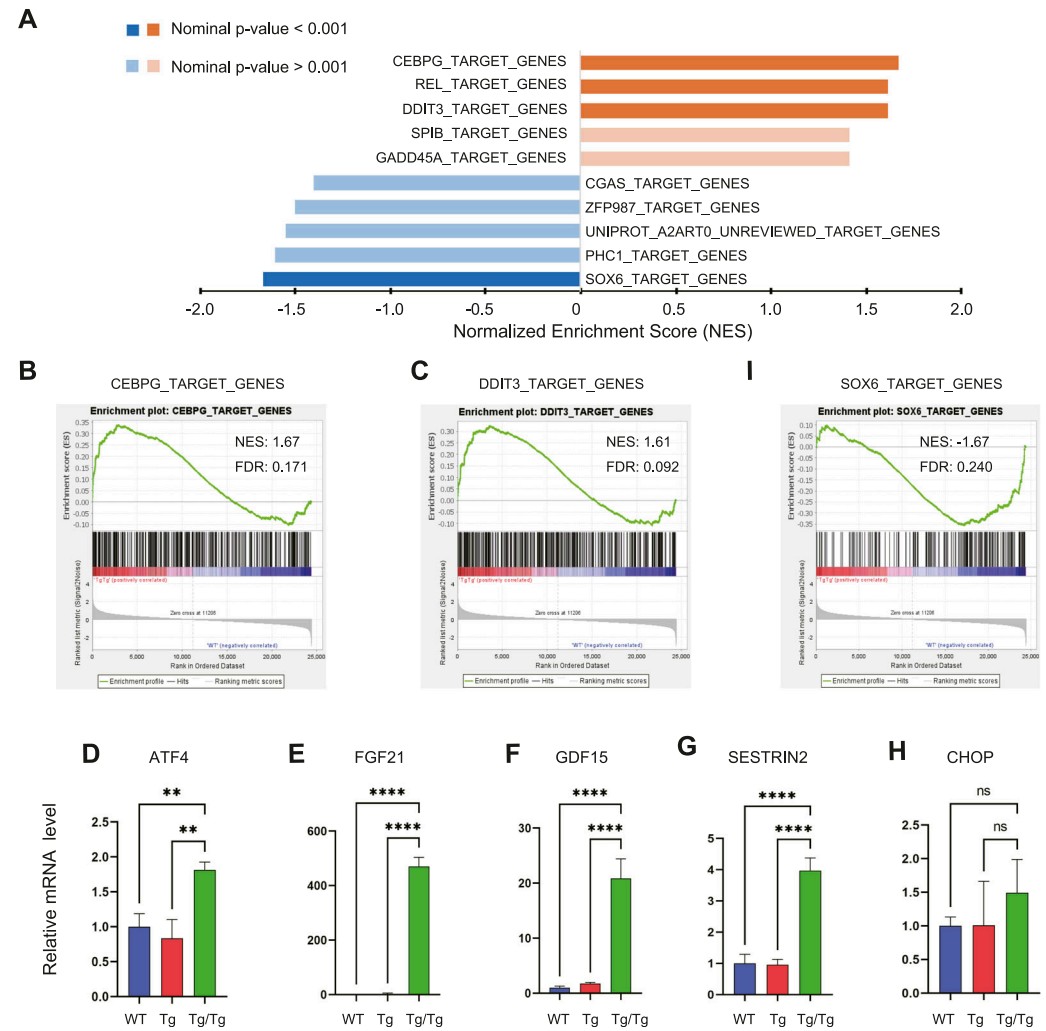

**Figure 2.   Comprehensive gene expression analysis by microarray (WT, n = 3; Tg/Tg, n = 3) and quantification of integrated stress response–related genes by qRT–PCR (WT, n = 4; Tg, n = 4; Tg/Tg, n = 4).**
**(A)** GSEA results with microarray measurements, using "M3: GTRD gene sets" as the database. When comparing WT and Tg/Tg, a positive (+) score reflects up-regulation by Tg/Tg, whereas a negative (−) score shows up-regulation in WT. See the "GSEA User Guide" for details on the parameters shown in the figure. **(B, C, I)** Enrichment plots by GSEA. NES, normalized enrichment score; FDR, false discovery rate. In GSEA, an FDR of less than 25% was considered an appropriate cutoff. **(D, E, F, G, H)** Results of qRT–PCR quantification of integrated stress response–related genes. Error bars are presented as the mean + SD of mice in each group. \*\*$P < 0.01$, \*\*\*\*$P < 0.0001$ versus Tg/Tg mice. One-way ANOVA with Sidak's multiple comparisons test.

the proteins and metabolites detected in the fatty acid degradation pathway in color. The proteins in this pathway that we detected tended to increase in Tg/Tg compared with WT and Tg (Fig 4B). ECHS1, HADHA, ACAA2, and ECI1 showed a significant increase in Tg/Tg. As for metabolites, an increase in acetyl-CoA was observed in Tg/Tg (Fig 4C). As metabolites involved in fatty acid degradation, we also analyzed acylcarnitines, which are converted to acyl-CoA upon passage through the mitochondrial inner membrane, and observed an increasing trend in Tg/Tg (Fig S4).

In the GSEA performed from the DNA microarray results, fatty acid degradation was increased in Tg/Tg mice (WT versus Tg/Tg: NES, 1.10; nominal *P*-value, 0.245; false discovery rate (FDR), 0.429. Tg versus Tg/Tg: NES, 1.71; nominal *P*-value < 0.001; FDR, 0.002; Fig 4D). The gene expression signals that contribute to these enrichment scores are shown in the heat map (Fig S5).

## Enhanced BCAA degradation pathway in the skeletal muscle of TFAM-overexpressing mice

Next, we proceeded to interpret the BCAA degradation pathway. Fig 5A shows the proteins and metabolites detected in the BCAA degradation pathway in color. The proteins in this pathway that we detected tended to increase in Tg/Tg compared with WT and Tg (Fig 5B). ECHS1, HADHA, and ACAA2 are increased because the same enzymes are used in this pathway as in the fatty acid degradation pathway, but in addition to these findings, there was also a significant increase in Tg/Tg, especially for BCKDHA, HIBADH, and HADHB. Acetyl-CoA is a terminal metabolite in this pathway, as in the fatty acid degradation pathway, but valine, leucine, and isoleucine show a decreasing trend in Tg/Tg, indicating that degradation is progressing (Fig 5C).

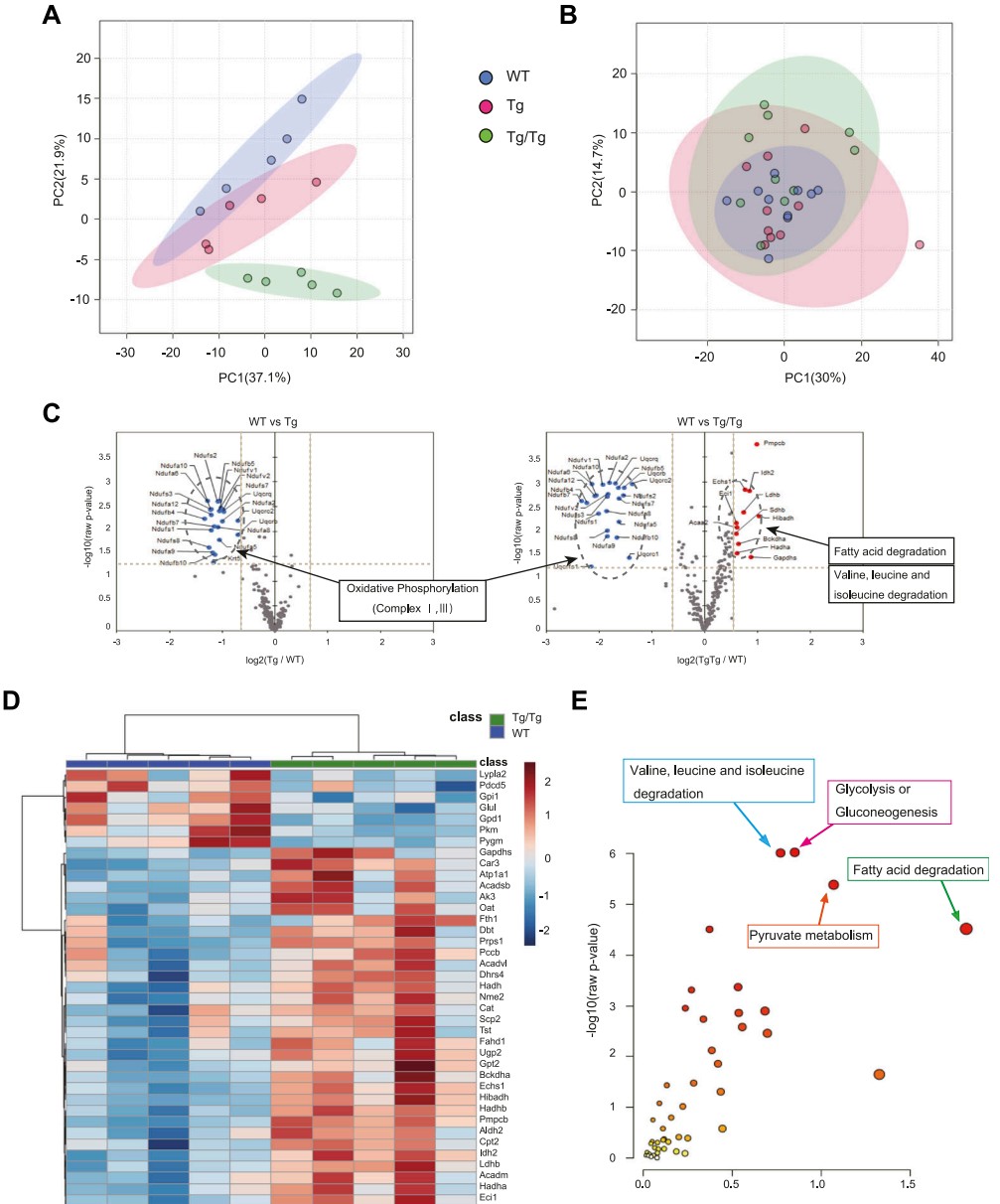

**Figure 3. Comprehensive analysis of the three types of mouse skeletal muscle using iMPAQT and metabolomics.**
**(A)** PCA plots with quantitative results for 247 proteins detected by iMPAQT (WT, n = 5; Tg, n = 5; Tg/Tg, n = 5). **(B)** PCA plots with quantitative results for 329 metabolites detected by metabolomics (WT, n = 10; Tg, n = 10; Tg/Tg, n = 10). **(C)** iMPAQT results are shown in a volcano plot. Proteins are shown that exceed the thresholds |fold change| > 1.5 and t test *P*-value < 0.05, with decreases shown in blue and increases shown in red. **(D)** Heat map showing proteins (excluding respiratory chain complexes) that satisfy t test *P* < 0.05 in the comparison between WT and Tg/Tg, which were extracted from the iMPAQT results. **(E)** Metabolic processes enriched by an integrative analysis of two parameters for the protein selected in **(D)** and the metabolite giving t test *P* < 0.05 for the comparison between WT and Tg/Tg. The analysis used MetaboAnalyst 5.0.

In the results of the GSEA performed from the DNA microarray results, BCAA degradation was increased in Tg/Tg mice (WT versus Tg/Tg: NES, 1.88; nominal *P*-value, 0.004; FDR, 0.050; Fig 5D. Tg versus Tg/Tg: NES, 1.94; nominal *P*-value, 0.002; FDR, 0.023). The gene expression signals that contribute to this enrichment are shown in the heat map (Fig S6). There was also a decrease in the expression of genes involved in myogenesis when comparing WT to Tg/Tg (NES, –1.20; nominal *P*-value, 0.079; FDR, 0.353; Fig 5E).

Using the GC-MS simultaneous analysis method, valine, leucine, and isoleucine were measured in the three types of mouse plasma (Fig 5F). A decreasing trend in Tg/Tg was also observed here, as seen for the muscle tissue, suggesting increased degradation in the muscle tissue.

## Overview of alterations in the central metabolic pathways and a metabolic shift in TFAM-overexpressing mice

Glycolysis and the central metabolic pathways that ranked highly in the integrated proteomics and metabolomics will be mentioned last. Fig 6A shows the proteins and metabolites detected in the glycolysis and tricarboxylic acid (TCA) cycle in color.

No significant changes in protein expression in glycolysis were observed in the three types of mice (Fig 6B), but all metabolites, from 2,3-bisphosphoglycerate to pyruvate, showed a significant decrease in Tg/Tg (Fig 6C). Randle et al found that fatty acid degradation reduces the activity of cytosolic glycolytic enzymes that precede pyruvate synthesis through a multistep process

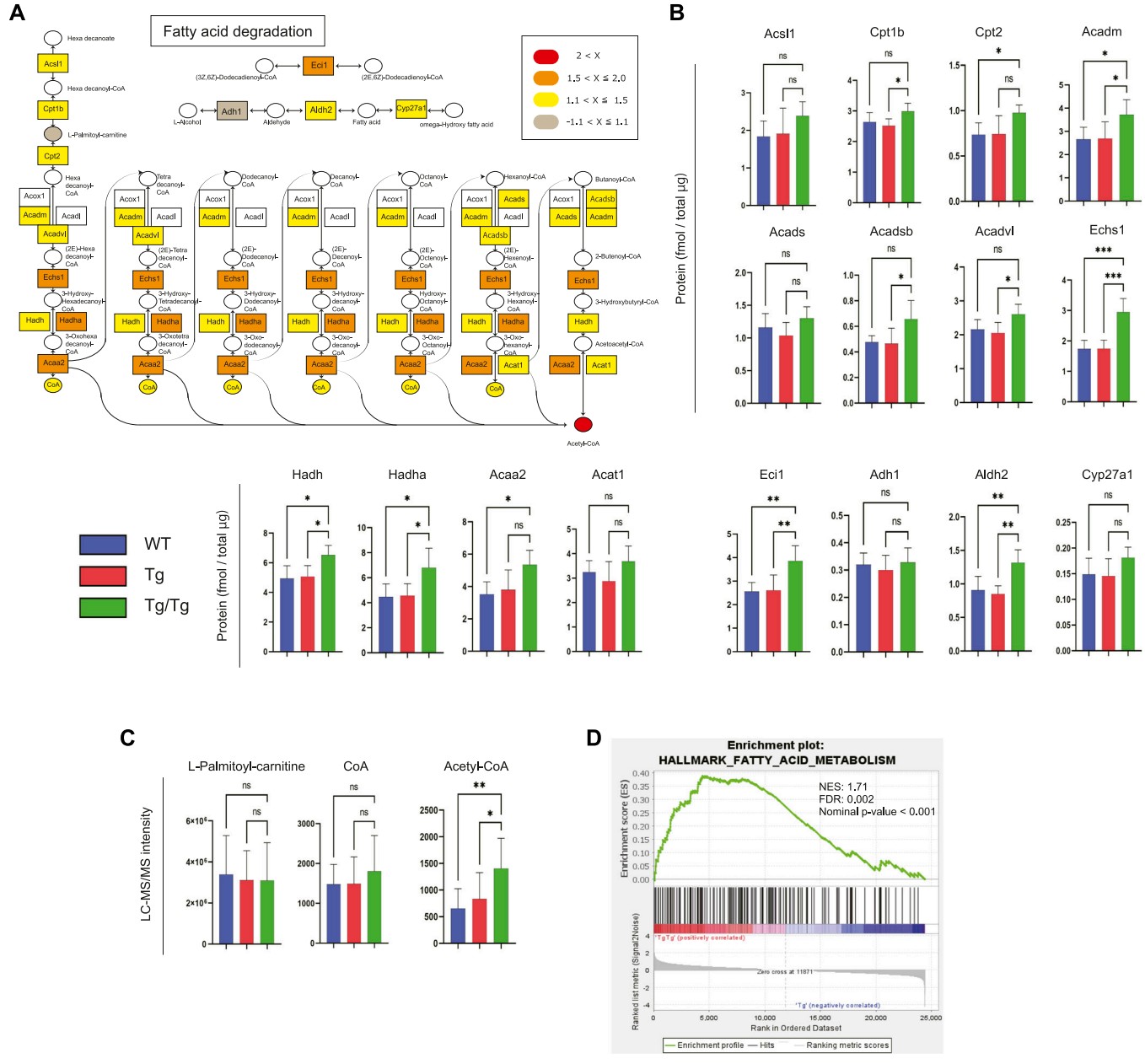

**Figure 4. Analysis of the fatty acid degradation pathway by proteomics and metabolomics.**
**(A)** Overview of proteins and metabolites detected in the fatty acid degradation pathway. The fold change was calculated from the average of the WT and Tg/Tg measurements, and the values are indicated by color coding. **(B)** Quantification of proteins extracted from the three types of mouse gastrocnemius muscle by the iMPAQT method (WT, n = 5; Tg, n = 5; Tg/Tg, n = 5). Error bars are presented as the mean + SD of mice in each group. *$P < 0.05$, **$P < 0.01$, ***$P < 0.001$ versus Tg/Tg mice. One-way ANOVA with Sidak's multiple comparisons test. **(C)** Quantification of metabolites extracted from the three types of mouse gastrocnemius muscle by metabolomics (WT, n = 10; Tg, n = 10; Tg/Tg, n = 10). Error bars are presented as the mean + SD of mice in each group. *$P < 0.05$, **$P < 0.01$ versus Tg/Tg mice. One-way ANOVA with Sidak's multiple comparisons test. **(D)** Enrichment plots by GSEA. NES, normalized enrichment score; FDR, false discovery rate. The results of Tg versus Tg/Tg are shown. "MH: hallmark gene sets" was used as the database. When comparing Tg and Tg/Tg, a positive (+) score reflects up-regulation in Tg/Tg, and a negative (–) score shows up-regulation in Tg.

involving mitochondrial and cytosolic reactions (Garland et al, 1968; Randle, 1998; De Oliveira & Liesa, 2020). This process is termed the Randle cycle and is attributed to an increase in acetyl-CoA because of fatty acid degradation. Although a decrease in the activity of pyruvate kinase and pyruvate dehydrogenase has been proposed under Randle cycle conditions, PKM and PDHA showed a slight decreasing trend in Tg/Tg mice in the present study (Fig 6B).

It is interesting to note that fatty acid degradation and BCAA degradation, which increase in Tg/Tg mice herein, result in an elevation of their product, acetyl-CoA, which shows very similar behavior to the Randle cycle.

In the TCA cycle, both proteins and metabolites were slightly increased in Tg/Tg mice compared with WT (Fig 6B and C). IDH2, SDHA, and SDHB showed a marked increase in expression in Tg/Tg mice.

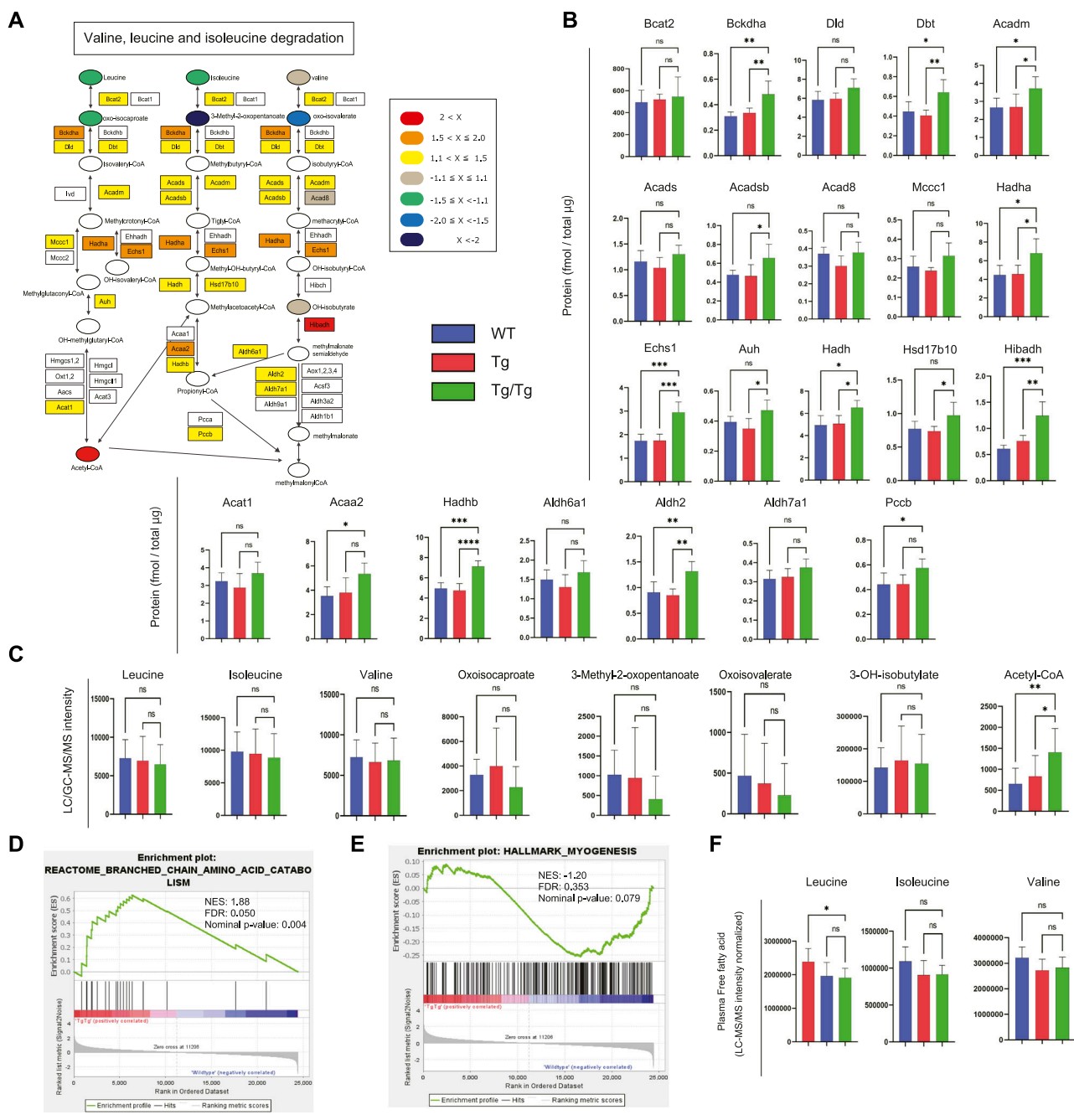

**Figure 5. Analysis of the BCAA degradation pathway by proteomics and metabolomics.**
**(A)** Overview of proteins and metabolites detected in the BCAA degradation pathway. The fold change was calculated from the average of the WT and Tg/Tg measurements, and the values are indicated by color coding. **(B)** Quantification of proteins extracted from the three types of mouse gastrocnemius muscle by the iMPAQT method (WT, n = 5; Tg, n = 5; Tg/Tg, n = 5). Error bars are presented as the mean + SD of mice in each group. *P < 0.05, **P < 0.01, ***P < 0.001, ****P < 0.0001 versus Tg/Tg mice. One-way ANOVA with Sidak's multiple comparisons test. **(C)** Quantification of metabolites extracted from the three types of mouse gastrocnemius muscle by metabolomics (WT, n = 10; Tg, n = 10; Tg/Tg, n = 10). Error bars are presented as the mean + SD of mice in each group. *P < 0.05, **P < 0.01 versus Tg/Tg mice. One-way ANOVA with Sidak's multiple comparisons test. **(D, E)** Enrichment plots by GSEA. NES, normalized enrichment score; FDR, false discovery rate. The results of WT versus Tg/Tg are shown. When comparing WT and Tg/Tg, a positive (+) score indicates up-regulation in Tg/Tg, whereas a negative (−) score means up-regulation in WT. **(D, E)** Databases used were "M2: Reactome subset of CP" (D) and "MH: hallmark gene sets" (E). **(F)** Quantification of BCAA extracted from the three types of mouse plasma by metabolomics (WT, n = 10; Tg, n = 10; Tg/Tg, n = 10). Error bars are presented as the mean + SD of mice in each group. *P < 0.05 versus Tg/Tg mice. One-way ANOVA with Sidak's multiple comparisons test.

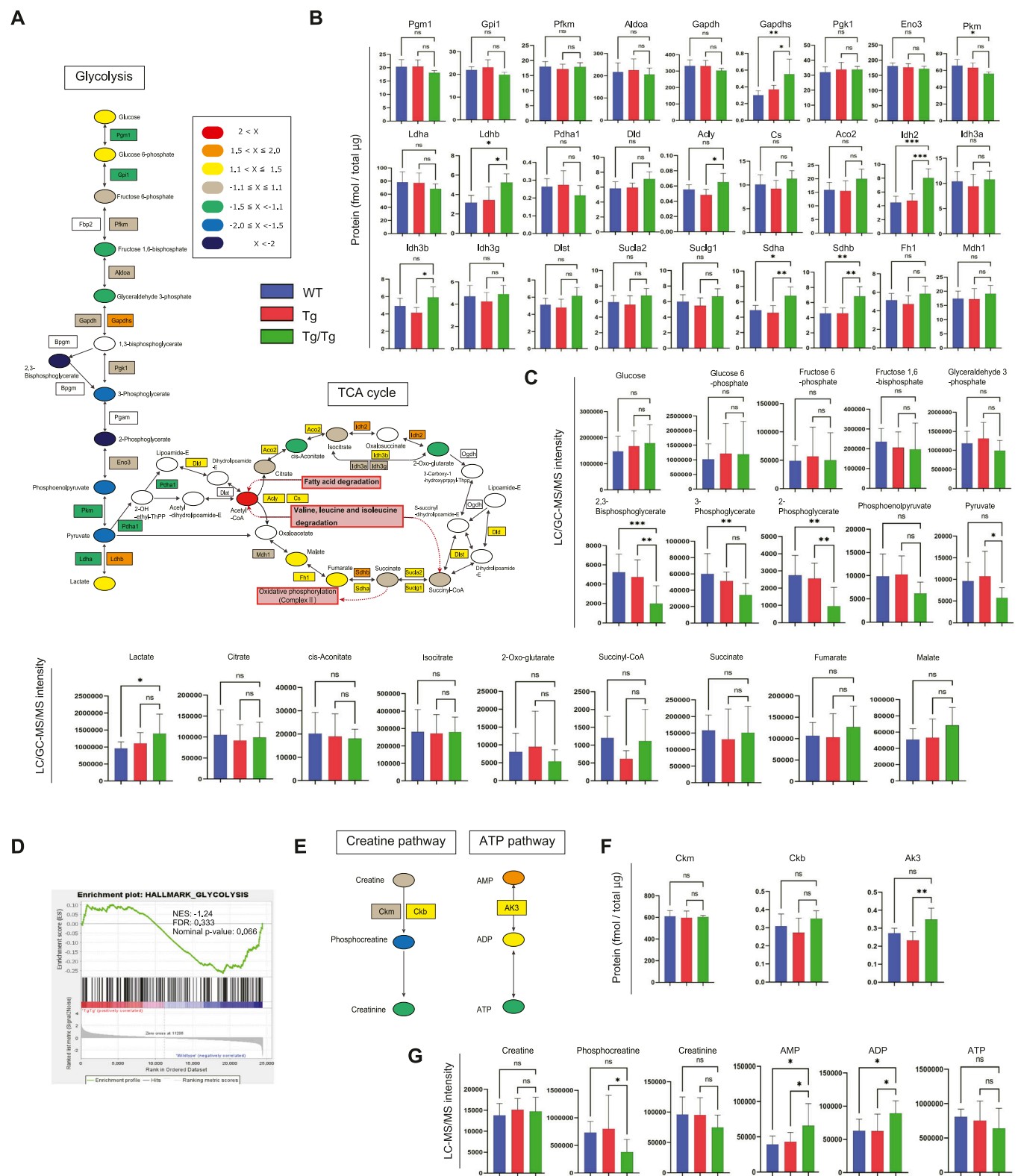

**Figure 6. Analysis of the glycolysis and the TCA cycle by proteomics and metabolomics.**
**(A)** Overview of proteins and metabolites detected in the glycolysis and the TCA cycle. The fold change was calculated from the average of the WT and Tg/Tg measurements, and the values are indicated by color coding. **(B)** Quantification of proteins extracted from the three types of mouse gastrocnemius muscle by the iMPAQT method (WT, n = 5; Tg, n = 5; Tg/Tg, n = 5). Error bars are presented as the mean + SD of mice in each group. *$P < 0.05$, **$P < 0.01$, ***$P < 0.001$ versus Tg/Tg mice. One-way ANOVA with Sidak's multiple comparisons test. **(C)** Quantification of metabolites extracted from the three types of mouse gastrocnemius muscle by metabolomics (WT,

In the GSEA performed from the DNA microarray results, glycolysis was decreased in Tg/Tg mice (WT versus Tg/Tg: NES, −1.24; nominal *P*-value, 0.066; FDR, 0.333; Fig 6D) and was not enriched in the comparison between Tg and Tg/Tg.

The mouse gastrocnemius muscle has both oxidative and glycolytic fiber muscle that uses glycolysis to provide energy (Jacobs et al, 2013; Rakus et al, 2015), but in Tg/Tg mice, the glycolytic function is thought to be attenuated compared with WT, so a compensatory pathway may be necessary. Phosphocreatine plays an important role in energy backup in skeletal muscle, and the proteins and metabolites involved in the creatine pathway detected in this study are shown in Fig 6E. In a comparison between the three types of mice, it was observed that the amount of phosphocreatine was decreased in Tg/Tg, although creatine and creatine kinase were unchanged (Fig 6F and G). In addition, a decreasing trend was observed for ATP, even though AMP and ADP were increasing at Tg/Tg mice (Fig 6G). This suggests that Tg/Tg mice may have a permanent shortage of energy, especially during spontaneous exercise.

# Discussion

In the present study, we have observed that transgenic mice expressing hTFAM, especially homozygous mice, have a unique phenotype, small and thin but with a progressive lifespan (Fig 1A–D). Because hTFAM expression is regulated under the *β*-actin promoter, we proceeded to analyze skeletal muscle, which is thought to be highly affected by TFAM enhancement. In our observations, the mtDNA-encoded gene and protein expression of respiratory chain complexes 1, 3, and 4 were decreased, especially in Tg/Tg. Our findings on respiratory chain complex expression on TFAM overexpression varied by tissue type. Ikeda et al reported decreased mtDNA-encoded genes in the left ventricle and aorta of TFAM-overexpressing mice (Ikeda et al, 2015), whereas Matsushima et al reported that TFAM overexpression tends to cause transcriptional repression in *Drosophila* Schneider cells (Matsushima et al, 2010). Recent reports have also shown that very high TFAM expression in mice suppresses mtDNA transcription (Bonekamp et al, 2021). In our observations here, TFAM overexpression may cause overpacking of the mtDNA and/or D-loops, as there is a more distinct decrease in Tg/Tg (Ohgaki et al, 2007). Subsequent down-regulation of these mitochondrial genes/proteins in Tg/Tg mice could be a source of stress signals. It is also undeniable that the excess of hTFAM as a protein may functionally compete with mouse TFAM and impair the original function of mouse TFAM. In such a case, however, mtDNA-derived mRNA would be uniformly affected, but this was not the case (Fig 1G).

In the comprehensive gene analysis using microarrays, genes involved in stress signaling were selected as the focus for enrichment analysis of genes with differential expression targeted by transcription factors (Fig 2A–C). C/EBP-γ, which forms heterodimers with ATF4 and regulates transcription, is reported to be involved in specific stress responses (Brearley-Sholto et al, 2021; Renfro et al, 2022) and has attracted attention as an autophagy regulator in amino acid starvation (Kim et al, 2022). In neural tissue, mitochondrial disruption has also been reported to activate an adaptive program known as the ISR (Yagi et al, 2017). The ISR causes reprogramming of global protein synthesis and triggers translation of specific mRNAs, such as Atf4 (Pakos-Zebrucka et al, 2016; Sasaki et al, 2020; Jiang et al, 2022). The overexpression of ATF4 in mouse skeletal muscle results in muscle fiber atrophy, and ATF4 increases the expression of downstream muscle atrophy–related genes (Ebert et al, 2010, 2012). In the present study, the qRT–PCR results also showed an increased expression of Atf4 in Tg/Tg mice. It is possible that stress-induced ATF4 acts directly on muscle atrophy. Genes targeted by DDIT3/CHOP, ATF4, FGF21, GDF15, and SESTRIN2 were increased in Tg/Tg mice, which is also representative of the mitochondrial stress response. The mitochondrial stress response is a biphasic hormesis, also termed mitohormesis (Yun & Finkel, 2014). Mild mitochondrial stress stimulates cytoprotective adaptations, such as the induction of responses in the nucleus and various cytoplasmic proteostasis maintenance pathways (D'Amico et al, 2017), whereas severe mitochondrial stress is detrimental. In recent years, mitohormesis has been attracting attention as a necessary component of lifespan extension (Owusu-Ansah et al, 2013; Burtscher et al, 2023). Systemic mitokines are thought to be one of the factors responsible for the observed lifespan progression (Burtscher et al, 2023), and herein, we observed very large increases in Fgf21 and Gdf15 gene expression in Tg/Tg mice. Induction of Fgf21 can produce metabolic responses in various organs, including increased browning of white adipose tissue (Kim et al, 2013), and Fgf21 and Gdf15 have differentially beneficial systemic consequences for mitochondrial stress responses (Kang et al, 2021). Although mitochondrial stress is present in this study, and an ISR is occurring, there may also be the possibility of proteotoxic stress resulting from the overexpression of foreign proteins.

The framework of this study was phenotypic analysis by proteomics and metabolomics. The iMPAQT results clearly reflect the differences between the three types of skeletal muscle, whereas the metabolomics showed high individual variability. Because metabolites reflect immediate conditions, tissues that are in a constant state of activity, such as skeletal muscle, are difficult to analyze. The iMPAQT results clearly show that the fatty acid and BCAA degradation pathways are enhanced in Tg/Tg mice, and the decline of Tg/Tg at the end of glycolysis is an interpretation recognized from the metabolomic results.

TFAM Tg/Tg mice are phenotypically small and thin, which appears to be closely related to mitochondrial stress and reductions in

---

n = 10; Tg, n = 10; Tg/Tg, n = 10). Error bars are presented as the mean + SD of mice in each group. *P < 0.05, **P < 0.01, ***P < 0.001 versus Tg/Tg mice. One-way ANOVA with Sidak's multiple comparisons test. **(D)** Enrichment plots by GSEA. NES, normalized enrichment score; FDR, false discovery rate. The results of WT versus Tg/Tg are shown. "MH: hallmark gene sets" was used as the database. When comparing WT and Tg/Tg, a + score shows up-regulation in Tg/Tg, and a − score indicates up-regulation in WT. **(E)** Overview of the proteins and metabolites detected in the creatine and ATP pathway. The fold change was calculated from the average of the WT and Tg/Tg measurements, and the values are indicated by color coding. **(F)** Quantification of proteins extracted from the three types of mouse gastrocnemius muscle by the iMPAQT method (WT, n = 5; Tg, n = 5; Tg/Tg, n = 5). Error bars are presented as the mean + SD of mice in each group. **(G)** Quantification of metabolites extracted from the three types of mouse gastrocnemius muscle by metabolomics (WT, n = 10; Tg, n = 10; Tg/Tg, n = 10). Error bars are presented as the mean + SD of mice in each group. *P < 0.05 versus Tg/Tg mice. One-way ANOVA with Sidak's multiple comparisons test.

mitochondrial proteins. If less ATP is produced in Tg/Tg mice than in WT, the smaller body size may be an adaptive strategy that can work more efficiently and expend less energy, contributing to energy conservation.

The observed enhancement of fatty acid degradation in Tg/Tg skeletal muscle may be an example of mitohormesis, and we believe that a specific enhancement of mitochondrial function occurs. As an example, *Echs1* deficiency is thought to cause OXPHOS reduction (Haack et al, 2015; Sharpe & McKenzie, 2018), but in contrast to this, we observed high ECHS1 expression in Tg/Tg. In general, moderate exercise uses fat as an energy source (Houten & Wanders, 2010; Muscella et al, 2020), and the relationship to lifespan progression is interesting. Fujii et al reported that respiratory exchange ratios were significantly lower in Tg/Tg mice under high-fat diet conditions (Fujii et al, 2022), consistent with accelerated fatty acid degradation.

Most amino acids of dietary origin are metabolized in the liver; however, BCAAs are often metabolized in tissues such as skeletal muscle (Blair et al, 2021). Thus, BCAAs are the only essential amino acids that provide a source of energy in muscles. In our proteomic analysis, we observed a very large up-regulation of Bckdha in Tg/Tg mice. Mice lacking BDK, an inactivator of BCKDH (BDK-KO mice), have been reported (Joshi et al, 2006). These mice show increased BCAA degradation, and their phenotype closely resembles that of our *TFAM* Tg/Tg mice, with smaller bodies and reduced mass of skeletal muscle and other organs (Joshi et al, 2006). The BCAA degradation pathway is activated throughout Tg/Tg mice in this study, which may be the key to the acquisition of the small body size. Amino acid starvation signals are said to up-regulate FGF21 expression in skeletal muscle and systemically promote fatty acid oxidation (Guridi et al, 2015; Shimizu et al, 2015). As a result of constant BCAA degradation activity, emaciated skeletal muscle may also exhibit amino acid starvation. The overexpression of FGF21 observed in Tg/Tg skeletal muscle may be due in part to these boosting effects.

The acetyl-CoA generated from the fatty acid and BCAA degradation that is activated in Tg/Tg mice may enter the TCA cycle and drive it permanently. This would seem to selectively inhibit the formation of acetyl-CoA from glucose in Tg/Tg mice, supported by the fact that all metabolites, from 2,3-bisphosphoglycerate to pyruvate, are reduced in Tg/Tg mice. We hypothesize that down-regulation of the glycolytic system would also be supported by a decrease in phosphocreatine, an instantaneous energy backup compound. This would be the full extent of the metabolic shift in the skeletal muscle of the Tg/Tg mice, which is consistent with the phenotype of a lean, small stature.

In this study, mice with stressed skeletal muscle mitochondria were observed to acquire a phenotype that is not pathological, but rather healthy, with progression of the lifespan. The key factors for this lifespan progression are newly identified herein as mitohormesis and a small body size because of increased fatty acid and BCAA degradation. Because the TFAM Tg/Tg mouse is a whole-body transgenic mouse, the extent to which the skeletal muscle phenotype observed in this study contributes to the lifespan-extending effect should be clarified in future studies. On the contrary, there is also the possibility of other biological mechanisms that can lead to this phenotype (e.g., autophagy changes promote the effect because of the accumulation of OXPHOS proteins), which needs to be verified. Recently, it has been reported that BCAA restriction in mice improves metabolic health and contributes to lifespan progression (Richardson et al, 2021; Trautman et al, 2022). The results of this study will provide clues regarding a healthy lifespan in the future.

# Materials and Methods

### TFAM Tg and Tg/Tg mice

We used transgenic mice expressing hTFAM that were originally generated in our laboratory by Ikeuchi et al (2005). TFAM transgene resided in the intron 1 of the Irak3 (IRAK-M) gene on chromosome 10 (Ylikallio et al, 2010). The protein levels of IRAK-M were not altered in the corresponding mouse models as determined by Western blotting (Fig 1E). Tg and Tg/Tg mice used in the analysis were 12-wk-old males. All procedures and animal care were approved by the Committee on Ethics of Animal Experiment for the Graduate School of Medical and Pharmaceutical Sciences at Kyushu University and were performed in accordance with the Guidelines for Animal Experiments of Kyushu University (A26–084, A28–084).

### Immunoblotting

Mouse muscle tissues were immediately frozen in liquid nitrogen. Tissues were lysed with RIPA buffer (50 mM Tris–HCl, pH 8.0, 150 mM NaCl, 0.5% sodium deoxycholate, 1% NP-40, 0.1% SDS, protease inhibitor cocktail [Wako]), homogenized by sonication, and then subjected to immunoblotting. Protein quantities were adjusted to 5 μg/lane using the BCA protein assay kit (Nacalai Tesque). Adjusted samples were separated using SDS–PAGE gels and transferred to PVDF membranes. The membranes were blocked using Blocking One (Nacalai Tesque) and then probed overnight with primary antibodies (IRAK-M: #I5157; Sigma-Aldrich, β-actin: #A5441; Sigma-Aldrich). The membranes were incubated with secondary antibodies. Proteins were detected by enhanced chemiluminescence (GE Healthcare). Chemiluminescence was recorded and quantified using a chilled charge-coupled device camera (LAS1000plus).

### Quantitative real-time PCR analyses

Total RNA was extracted from the gastrocnemius muscle using RNeasy Mini Kit (QIAGEN) in accordance with the manufacturer's instructions. RNA concentration (ng/ml) and sample purity (the 260/280 ratio) were measured using a NanoDrop 1000 spectrophotometer (Thermo Fisher Scientific). Reverse transcription of total RNA was performed with PrimeScript RT Reagent Kit (Takara), in accordance with the manufacturer's protocol. Gene expression was measured by SYBR Green–based qRT–PCR. Relative quantification was performed using the comparative cycle threshold (Ct) method, relative to 18S ribosomal RNA. The PCR primers are listed in Table 1.

### iMPAQT method (proteomics)

First, the internal standards used in the iMPAQT method were prepared as follows: for each of the 1,123 metabolic enzymes, at

**Table 1. List of primer sequences used for qRT–PCR analysis.**

| Primer name | Sequence 5'→3' |
| --- | --- |
| 18S Forward | CGCGGTTCTATTTTGTTGGT |
| 18S Reverse | AGTCGGCATCGTTTATGGTC |
| mouseTFAM Forward | AAGGATGATTCGGCTCAGG |
| mouseTFAM Reverse | GGCTTTGAGACCTAACTGG |
| humanTFAM Forward | ATGGCGTTTCTCCGAAGCAT |
| humanTFAM Reverse | TCCGCCCTATAAGCATCTTGA |
| 12S Forward | CCGCTCTACCTCACCATCTC |
| 12S Reverse | CCCATTTCATTGGCTACACC |
| 16S Forward | GGGATAACAGCGCAATCCTA |
| 16S Reverse | GATTGCTCCGGTCTGAACTC |
| ND1 Forward | GGATCCGAGCATCTTATCCA |
| ND1 Reverse | GGTGGTACTCCCGCTGTAAA |
| ND2 Forward | AGGGATCCCACTGCACATAG |
| ND2 Reverse | CTCCTCATGCCCCTATGAAA |
| COX1 Forward | GGTCAACCAGGTGCACTTTT |
| COX1 Reverse | TGGGGCTCCGATTATTAGTG |
| COX2 Forward | ACGAAATCAACAACCCCGTA |
| COX2 Reverse | GGCAGAACGACTCGGTTATC |
| ATP8 Forward | AACATTCCCACTGGCACCT |
| ATP8 Reverse | GGGGTAATGAATGAGGCAAA |
| ATP6 Forward | CCTTCCACAAGGAACTCCAA |
| ATP6 Reverse | GGTAGCTGTTGGTGGGCTAA |
| COX3 Forward | CAAGGCCACCACACTCCTAT |
| COX3 Reverse | ATTCCTGTTGGAGGTCAGCA |
| ND3 Forward | TTCGACCCTACAAGCTCTGC |
| ND3 Reverse | TGAATTGCTCATGGTAGTGGA |
| ND4L Forward | TTCTTCAACCTCACCATAGCC |
| ND4L Reverse | GGCTGCGAAAACTAAGATGG |
| ND4 Forward | CCACTGCTAATTGCCCTCAT |
| ND4 Reverse | CTTCAACATGGGCTTTTGGT |
| ND5 Forward | TCCTACTGGTCCGATTCCAC |
| ND5 Reverse | TTTGATGTCGTTTTGGGTGA |
| ND6 Forward | CGATCCACCAAACCCTAAAA |
| ND6 Reverse | TTGGTTGTCTTGGGTTAGCA |
| Cytb Forward | TGAGGGGGCTTCTCAGTAGA |
| Cytb Reverse | CTGTTTCGTGGAGGAAGAGG |
| ATF4 Forward | TCGATGCTCTGTTTCGAATG |
| ATF4 Reverse | AGAATGTAAAGGGGGCAACC |
| FGF21 Forward | GGGAGGATGGAACAGTGGTA |
| FGF21 Reverse | GTCCTCCAGCAGCAGTTCTC |
| GDF15 Forward | CTTGAAGACTTGGGCTGGAG |
| GDF15 Reverse | TAAGAACCACCGGGGTGTAG |
| SESTRIN2 Forward | TAGCCTGCAGCCTCACCTAT |

**Table 1. Continued**

| Primer name | Sequence 5'→3' |
| --- | --- |
| SESTRIN2 Reverse | CTACGGGTCGTCTTCTCAGG |
| CHOP Forward | CAGAGGTCACACGCACATCC |
| CHOP Reverse | CCTTGCTCTTCCTCCTCTTCC |

least one proteotypic peptide was selected. 24 QconCAT (Scott et al, 2016) sequences were designed by linking ~50 proteotypic peptides each. The pENTR2B vector (Thermo Fisher Scientific) containing the designed QconCAT sequences was produced at Eurofins Genomics (Supplemental Data 1) and was used to recombine with the pET21b-QcodeAA-DEST vector to generate expression vectors. To synthesize a stable isotope QconCAT with maximum isotope labeling efficiency, the expression vector was introduced into *E. coli* (BL21 [DE3]), in which stable isotopes were inserted for arginine and lysine, to generate expression strains. Inclusion bodies were purified using Ni–resin (Probond, Invitrogen). The internal standard solution was prepared by mixing 24 different QconCAT solutions. The skeletal muscle tissue was frozen in liquid nitrogen and immediately crushed using a bead shaker (MB1200; Yasui Kikai). Lysis buffer (3.5 M urea, 1% SDS, 50 mM Tris–HCl) was added to the crushed tissue, and sonication was performed 10 times at 30-s intervals using an ultrasonic crusher (Sonic Bio, Bioruptor). The total protein concentration was then determined with Pierce BCA Protein Assay Kit (Thermo Fisher Scientific). To a sample aliquot containing 100 $\mu$g of total protein, an internal standard solution was added and treated with 10 mM Tris (2-carboxyethyl) phosphine hydrochloride (Thermo Fisher Scientific) at 37°C for 45 min followed by 20 mM 2-iodoacetamide (Sigma-Aldrich) at room temperature for 30 min, to reduce and alkylate the cysteine residues. The samples were then mixed with five volumes of acetone, allowed to stand at –30°C for 1 h, and centrifuged to obtain protein pellets. The pellet was washed twice with ice-cold 90% acetone, resuspended in 100 $\mu$l of digestion buffer (50 mM triethylammonium bicarbonate), and sonicated under the same conditions as tissue lysis. Each sample was digested with lysyl endopeptidase (1 $\mu$g; Wako) at 37°C for 1 h and further digested with trypsin (2 $\mu$g; Roche) at 37°C for 14 h. Trifluoroacetic acid was added to the digested material to a final concentration of 1% (vol/vol), centrifuged, and transferred to vials for analysis, of which 10 $\mu$l was introduced into liquid chromatography with tandem mass spectrometry (LC-MS/MS). MRM analysis was performed in conjunction with the ACQUITY UPLC H-Class (Waters) liquid QTRAP 6500 (SCIEX) triple quadrupole mass spectrometer in conjunction with a chromatography system. The main mass spectrometer parameters were as follows: ion spray voltage, 5,500 V; curtain gas, 30 psi; collision gas, 12; ion source gas 1, 40 psi; ion source gas 2, 60 psi; and source temperature, 350°C. The declustering potential (DP) was calculated according to the following equation: DP = (0.049 × Q1) + 42.6. The collision energy (CE) was calculated according to the following two equations: CE = ([0.036 × Q1] + 6.9) or CE = ([0.054 × Q1] – 2.4) (double-charged and corresponding to doubly and triply charged precursor ions, respectively), where Q1 is the mass/charge ratio of the precursor ion. The collision cell exit

potential (CXP) was calculated according to the following equation: CXP = (0.0391 × Q3) – 2.23, where Q3 is the mass/charge ratio of the fragment ion. The entrance potential (EP) was set to 10, and the resolution of Q1 and Q3 was set to "unit" (half-width, 0.7 D). All data acquisition was performed using the Scheduled MRM option, with the target scan time set to 1 s and the MRM detection window set to 120 s. The obtained mass spectrometry data were imported into Skyline Software (University of Washington), and peak picking was performed. Each peptide was monitored for three or more transitions, and transitions where interference because of foreign material was observed were eliminated. Quantitation values were calculated for peaks with matched peak retention times and a Ratio Dot Product greater than 0.9. The peak area ratio of endogenous peptides to ($^{13}C_6/^{15}N_4$) Arg and ($^{13}C_6/^{15}N_2$) Lys-labeled peptides (internal standard) was multiplied by the added QconCAT concentration to calculate the quantitative values. The artificial protein sequences and corresponding MRM assays are described in Supplemental Data 1.

### Gene expression microarrays

Total RNA was isolated from the gastrocnemius muscle tissue using TRIzol Reagent (Invitrogen Corp., Life Technologies) and then purified using the SV Total RNA Isolation and RNeasy System (Promega Corp.) according to the manufacturer's instructions. RNA samples were quantified using a NanoDrop ND-1000 spectrophotometer (Thermo Fisher Scientific, Inc.), and the quality of the RNA was confirmed with an Experion automated electrophoresis station (Bio-Rad Laboratories, Inc.). The complementary RNA was amplified and labeled by Low Input Quick Amp Labeling Kit (Agilent Technologies) according to the manufacturer's instructions, and hybridized to SurePrint G3 Human Gene Expression Microarrays 8 × 60 K (Agilent Technologies), a DNA chip including 60,000 genes. All hybridized microarray slides were scanned using an Agilent scanner. Relative hybridization intensities and background hybridization values were calculated using Agilent Feature Extraction Software. Pretreatment and microarrays were performed by an external service provider (Cell Innovator).

### Metabolomics

#### *Extraction of metabolites*

The gastrocnemius muscle tissue was immediately frozen in liquid nitrogen after organ removal and then broken into small chunks. These chunks were placed in small metal cone tubes, with the weight targeted around 50 mg. The chunks were crushed with a multibead shocker (Yasui Kikai; 2,000 rpm for 10 s) using a metal cone chilled with sufficient liquid nitrogen. After crushing, it was checked that the contents were in powder form, and they were then allowed to stand at room temperature for 30 s. Then, 0.7 ml of ice-cold 90% MeOH was added. After transferring the sample to a 2-ml tube, another 0.7 ml of ice-cold 90% MeOH was added, vortexed, sonicated five times (30 s of sonication and 30 s of cooling) using BIORUPTOR II, and centrifuged at 21,500$g$ for 5 min at 4°C. The supernatants were collected. Acylcarnitine analysis using

hydrophilic interaction liquid chromatography (HILIC) was used for the LC-MS analysis.

Ice-cold chloroform and ice-cold deionized water were further added to the supernatant at a solution ratio of MeOH:chloroform:water = 1:1:0.9, vortexed, and centrifuged at 21,500$g$ for 5 min at 4°C. The upper water/methanol phase was collected and dried using a miVac DUO concentrator system. It was then suspended in the starting mobile phase in LC-MS and subjected to analysis.

For plasma samples, plasma and MeOH were mixed at a ratio of 1:9. Sonication and centrifugation were then performed under the same conditions as above. Ice-cold chloroform and ice-cold water were added to the supernatant at the same ratio as above, and after centrifugation, the upper methanol/water phase was collected and dried.

### LC–MS analysis of metabolites

Skeletal muscle–derived metabolites were analyzed by LC–MS based on both the reversed-phase ion-pair chromatography and HILIC modes, coupled with a triple quadrupole mass spectrometer LCMS-8040 (Shimadzu), as described previously (Saito et al, 2017). For monitoring 60 kinds of metabolites, including intermediates, in central metabolism, reversed-phase ion-pair chromatography was performed using an ACQUITY UPLC BEH C18 column (100 × 2.1 mm, 1.7 $\mu$m particle size; Waters). Metabolites detected by this ion-pair method are appended with RP ion-pair in the name and are listed in Supplemental Data 1. The mobile phase consisted of solvent A (15 mM acetic acid and 10 mM tributylamine) and solvent B (methanol), and the column oven temperature was 40°C. The gradient elution program was as follows, with a flow rate of 0.3 ml/min: 0–3 min, 0% B; 3–5 min, 0–40% B; 5–7 min, 40–100% B; 7–10 min, 100% B; 10.1–14 min, 0% B. The parameters for negative electrospray ionization (ESI) mode under multiple reaction monitoring (MRM) were as follows: drying gas flow rate, 15 l/min; nebulizer gas flow rate, 3 l/min; DL temperature, 250°C; heat block temperature, 400°C; and CE, 230 kPa. For monitoring 51 kinds of metabolites, including amino acids, HILIC chromatography was performed using a Luna 3 $\mu$m HILIC 200A column (150 × 2 mm, 3 $\mu$m particle size; Phenomenex). Metabolites detected by the HILIC method are appended with HILIC in the name and are listed in Supplemental Data 1. The mobile phase consisted of solvent A (10 mM ammonium formate in water) and solvent B (9:1 of acetonitrile:10 mM ammonium formate in water), and the column oven temperature was 40°C. The gradient elution program was as follows, with a flow rate of 0.3 ml/min: 0–2.5 min, 100% B; 2.5–4 min, 100–50% B; 4–7.5 min, 50–5% B; 7.5–10 min, 5% B; 10.1–12.5 min, 100% B. The parameters for the positive and negative ESI modes under MRM were as described above. To monitor 29 kinds of acylcarnitine, the same HILIC chromatography method as above was used. Acylcarnitines detected by this method are listed in Supplemental Data 1. The parameters for the heated ESI in the positive ion mode under the precursor ion scan were as above. The acylcarnitine profiles were detected under a precursor ion scan for m/z 85.05 by changing the collision energy according to the length of the fatty acids: –20 for short-chain acylcarnitines (C0–C8); –25 for middle-chain

acylcarnitines (C10:1–C14:OH); and −35 for long-chain acylcarnitines (C16–C18:OH). LC-MS was also used for metabolites derived from mouse plasma using the same methods described above. Data processing was performed using the LabSolutions LC-MS software program (Shimadzu).

### Gas chromatography–mass spectrometry (GC-MS) analysis of metabolites

Methoxyamine hydrochloride pyridine solution (100 $\mu$l, 20 mg/ml) was added to the samples dried as above and incubated at 30°C for 90 min for methoxylation. MSTFA (50 $\mu$l) was then added to the solution, further incubated at 37°C for 30 min for trimethylsilylation (TMS), and subjected to GC-MS measurements.

A GCMS-TQ8030 gas chromatograph–tandem mass spectrometer (Shimadzu) was used for metabolomics. A DB-5 capillary column (30 m × 0.250 mm i.d.; 1.00-$\mu$m thickness; Agilent Technologies) was used for the chromatographic separation. The column oven temperature was initially held at 100°C for 4 min, then raised at 10°C/min to 320°C, with a final hold for 11 min. The interface temperature and the carrier gas (He) flow rate were set at 280°C and 1.1 ml/min, respectively. The mass spectrometer operating parameters were set as follows: ion source temperature, 200°C; electron energy, 70 eV. The MRM transitions were set according to a commercially available GC-MS Metabolite Database (Ver. 3.0; Shimadzu, Method file "OA_TMS_DB5_37min_V3_MRM"). The list of metabolites measured in this study is included in Supplemental Data 1. Samples were automatically injected in the splitless mode, and the injection volume was set at 1 $\mu$l. The sample injection sequence was randomly arranged.

### GSEA

GSEA (Subramanian et al, 2005) was used to interpret the results of comprehensive gene expression analysis by microarray in WT, Tg, and Tg/Tg mouse skeletal muscle. The GSEA software version was 4.3.2, and the parameters of the gene set parameters and run enrichment tests were set as follows. The name of each gene was selected as the expression dataset, and "m3.gtrd.v2023.1.Mm.symbols.gmt," "mh.all.v2023.1.Mm.symbols.gmt," and "m2.cp.reactome.v2023.1.Mm.symbols.gmt" were chosen as the gene set databases. The permutation value was set to the default value of 1,000 for computing the normalized enrichment score (NES). No_collapse was set, and the original dataset was used. The permutation type selected was "gene set." The chip platform used was "Mouse_AGILENT_Array_MSigDB.v2023.1.Mm.chip." Nominal *P*-values < 0.05, NES >1, and an FDR of q < 0.25 were considered statistically significant.

### Proteomic and metabolomic data analysis

Visualizations of the iMPAQT and metabolomic data were performed using MetaboAnalyst 5.0 to identify specific proteins and metabolites in TFAM-overexpressing mice, including PCA, heat maps, and integrated pathway analysis. The proteins and metabolites for each pathway ranked in the integrated analysis are shown in Supplemental Data 1.

### Statistical analysis

GraphPad Prism 9.2.0 (GraphPad Software) was used for statistical analysis. Relationships between the groups were compared using the log-rank Mantel–Cox test, two-way ANOVA with Tukey's multiple comparisons test, one-way ANOVA with Sidak's multiple comparisons test, and the Mann–Whitney multiple comparisons test.

## Data Availability

Additional data are provided as Supplemental Data and Figures. The MS proteomics data have been deposited to the PanoramaWeb (https://panoramaweb.org) with the dataset Permanent Link: https://panoramaweb.org/GqfOgy.url.

## Supplementary Information

## Acknowledgements

We would like to thank Dr. Daiki Setoyama for his guidance in obtaining the metabolomic data. We also thank the members of the Department of Clinical Chemistry and Laboratory Medicine, Kyushu University, for breeding TFAM transgenic mice and helpful discussions. We are grateful to LSI Medience Corporation for its generous support during this study. We thank Catherine Perfect, MA (Cantab), from Edanz (https://jp.edanz.com/ac), for editing a draft of this article. This research was supported by Grants-in-Aid for Scientific Research from the Japan Society for the Promotion of Science (grant numbers 17H01550 and 20H00530).

### Author Contributions

K Igami: conceptualization, data curation, formal analysis, investigation, methodology, and writing—original draft, review, and editing.
H Kittaka: data curation, visualization, and methodology.
M Yagi: resources and data curation.
K Gotoh: resources and data curation.
Y Matsushima: data curation.
T Ide: resources.
M Ikeda: resources.
S Ueda: data curation.
S-I Nitta: methodology.
M Hayakawa: validation and visualization.
KI Nakayama: methodology.
M Matsumoto: methodology.
D Kang: conceptualization, funding acquisition, and project administration.
T Uchiumi: conceptualization and supervision.

## Conflict of Interest Statement

K Igami, H Kittaka, and S-I Nitta are full-time employees of LSI Medience Corporation. K Igami, H Kittaka, and S-I Nitta are also seconded to and belong to Kyushu Pro Search Limited Liability Partnership, a subsidiary of LSI Medience Corporation. M Hayakawa holds employment with Kyushu Pro Search Limited Liability Partnership. We would like to declare these affiliations as potential conflict of interests. No disclosures were reported by the other authors.

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
