## [Reviewer comments · Life Science Alliance]

Life Science Alliance

iMPAQT reveals that adequate mitohormesis from TFAM overexpression leads to life extension in mice

Ko Igami, Hiroki Kittaka, Mikako Yagi, Kazuhito Gotoh, Yuichi Matsushima, Tomomi Ide, Masataka Ikeda, Saori Ueda, Shin-ichiro Nitta, Manami Hayakawa, Keiichi I Nakayama, Masaki Matsumoto, Dongchon Kang, and Takeshi Uchiumi

DOI: <https://doi.org/10.26508/lsa.202302498>

Corresponding author(s): Ko Igami, LSI Medience corp. and Takeshi Uchiumi, Kyushu University

Review Timeline:

Submission Date:	2023-11-28
Editorial Decision:	2024-01-16
Revision Received:	2024-03-24
Editorial Decision:	2024-04-09
Revision Received:	2024-04-13
Accepted:	2024-04-16

Transaction Report:

January 16, 2024

Re: Life Science Alliance manuscript #LSA-2023-02498-T

Dr. Ko Igami
LSI Medience corp.
2-3, Shibaura 1-chome
Minato-ku, Tokyo 105-0023
Japan

Dear Dr. Igami,

Thank you for submitting your manuscript entitled "iMPAQT reveals that adequate mitohormesis from TFAM overexpression leads to life extension in mice" to Life Science Alliance. The manuscript was assessed by expert reviewers, whose comments are appended to this letter. We invite you to submit a revised manuscript addressing the Reviewer comments.

Thank you for this interesting contribution to Life Science Alliance. We are looking forward to receiving your revised manuscript.

Sincerely,

B. MANUSCRIPT ORGANIZATION AND FORMATTING:

Reviewer #1 (Comments to the Authors (Required)):

Igami and colleagues show that mice carrying at least two copies of a transgenic cassette for human TFAM, seemingly one in each chromosome pair, show an extension in lifespan and a decrease in body weight. The authors characterize the muscle and plasma of these mice by transcriptomics, metabolomics, and proteomics to identify a signature that is compatible with the activation of the mitochondrial integrated stress response and an increase in fatty acid and amino acid oxidation. From these data, authors conclude that mitohormesis is induced by high levels of human TFAM expression, inducing metabolic shifts in muscle to explain life span extension. However, there are many technical aspects about this mouse model that are unclear, which impede drawing the conclusions stated by the authors. In addition, other functional analyses key to confirm their conclusions are missing. The major concerns are:

1) Both heterozygous and homozygous TFAM mice are defined in the text as TFAM overexpressors, a definition that is confusing and inaccurate. The reason is that Human TFAM is transgenically expressed and not mouse TFAM, so technically it is not an overexpression. To establish whether the phenotype observed is truly induced by differences in TFAM activity, the chromosomal loci where the TFAM transgenic cassettes are located must be defined, to validate that no endogenous genes are disrupted. The fact that the differences are only significant in Tg/Tg mice supports that the disruption of two endogenous alleles by the TFAM cassette, rather than a threshold in TFAM activity, explains the phenotype.

2) Related to point number 1, it is a possibility that the mitochondrial stress responses observed could be achieved by overexpressing any other human or non-mouse protein (even an inactive form of TFAM), with the phenotype observed is a consequence of proteotoxic stress that results from overexpressing a foreign protein. The lack of a strong phenotype in mice with the human TFAM transgene present only in one chromosome supports this possibility.

3) Human TFAM expressed at very high levels could even behave as a dominant negative form of mouse TFAM. Thus, other mechanisms, rather than over-packing due to supra-physiological TFAM activity as concluded by the authors, can explain the effects observed in TFAM tg/tg mice on mitochondrial gene expression.

4) Authors claim that an increase in fatty acid oxidation in muscle can explain the decrease in body weight. However, no measurements of whole-body energy expenditure, RER and mitochondrial fatty acid oxidation capacity in muscle (or other tissues such as liver or adipose) are shown, which are needed to draw this conclusion of increased fatty acid expenditure.

5) Related to point number 4, the authors conclude that a more efficient use of fatty acids for ATP production explains the decrease in FFA content in plasma. However, a more efficient use would have the opposite effect on circulating fatty acid levels: if transgenic mice produce ATP more efficiently from fatty acid oxidation, these mice will utilize less fatty acids to cover the same ATP demand and thus circulating FFA content should not change or even increase. An example is fasting, where fatty acid oxidation is increased in muscle and liver, but circulating levels are not decreased. In this regard, TFAM overexpression in adipose tissue could impair lipolysis and white adipose tissue function to decrease circulating fatty acid levels.

6) Do the authors think that the elevation of fatty acid oxidation is a consequence of the increase in FGF21 or just because higher TFAM content in mitochondria induces a direct effect on them to elevate their fatty acid oxidation capacity? Given that the TFAM tg/tg mouse line is a whole-body transgenic mouse, how much of the lifespan extension stems from human TFAM actions in the muscle, versus human TFAM actions in other tissues? These issues should be discussed in the manuscript, as well as toning down the conclusions on the contributions of TFAM actions in muscle to the phenotype.

7) Gastrocnemius is not a fast twitch muscle, it has both oxidative and glycolytic fibers.

8) Authors suggest that TFAM tg/tg might show muscle atrophy, given the signature of increased amino acid oxidation and decreased protein translation characteristic of ISR activation. How do authors reconcile these findings with increased lifespan? Would this increase in lifespan be meaningful for a wild mouse if the transgenic mouse cannot run fast and escape its predators?

9) Authors show a decrease in the mRNA content of different OXPHOS components, while proteomics data reveal an

accumulation of some OXPHOS proteins that showed a decrease in their mRNA levels. Authors conclude that this discrepancy between mRNA and protein can reflect mitohormesis and the ISR activation. However, such an increase in protein content in samples in which the ISR is activated, which decreases protein translation, will most likely be explained by a selective blockage of the degradation of some mitochondrial proteins. In addition, macroautophagy can also increase amino acid and fatty acid catabolism, with changes in mitochondrial shape being demonstrated to spare mitochondria from degradation when autophagy is activated. Thus, authors should either block the ISR with ISRIB or block its metabolic effector FGF21, GDF15, to test the dependency of the phenotype on the ISR, or measure autophagy and mitochondrial shape, to determine whether those could explain the phenotype.

Reviewer #2 (Comments to the Authors (Required)):

The paper addresses a unique role of a mitochondrial transcription factor TFAM in mitohormesis and resulting healthier phenotype. The authors performed the gene expression analysis which revealed mitochondrial stress in TFAM-overexpressing skeletal muscle leading to a beneficial adaptive response and life extension in mice. Proteomics and metabolomics were employed to demonstrate that overexpressing the TFAM gene in mice induced a metabolic shift towards enhanced degradation of fatty acids (FA) and branched-chain amino acids (BCAA) and decreased glycolysis. The presented data justified the conclusions of the paper but the following revisions should be considered.

1. While the iMPAQT method provides quantitative protein analysis, the LC-MS and GC-MS analyses of metabolites are qualitative. The targeted, quantitative LC-MS/MS or GC-MS analyses should be used to confirm that major metabolic intermediates of TCA cycle and FA and BCAA metabolism pathways are affected by the TFAM overexpression. In particular, branched-chain keto acids should be also analyzed in addition to BCAA to prove the increased degradation of BCAA in Tg/Tg mice.
2. The paper states enhanced fatty acid degradation pathway in skeletal muscle of TFAM-overexpressing mice. How do the authors explain the fact that while the increase of acetyl CoA was observed in Tg/Tg mice, acetyl carnitine showed no change?
3. The mass spec analysis showed decreased levels of FFA in Tg/Tg plasma and increased levels of many hydroxylated acylcarnitine species in Tg/Tg skeletal muscle, however the more abundant acylcarnitines such as palmitoyl carnitine and oleoyl carnitine that come from fatty acid oxidation showed no change. How does this agree with the statement that the fatty oxidation pathway is enhanced by the TFAM overexpression?
4. The authors suggest that Tg/Tg mice may have a permanent shortage of energy. To test this hypothesis, ATP and AMP levels should be analyzed in the skeletal muscle.
5. The authors state that BCAAs are rarely metabolized in the liver and are instead metabolized in tissues, primarily skeletal muscle. This statement needs to be double checked as there are many published reports showing BCAA metabolism in liver.

Reviewer #3 (Comments to the Authors (Required)):

Igami et al. present a comprehensive omics analysis of TFAM overexpressing mice strains (Tg and Tg/Tg) and identify respective metabolic adaptations. Using their iMPAQT pipeline for absolute protein quantification published elsewhere they report that TFAM overexpression results in mitohormesis, increased FA and AA degradation and decreased glycolysis in skeletal muscle predominantly. They furthermore integrate and crosscorrelate metabolomic data and use DNA microarray gene expression and report typical mitochondrial stress response markers (e.g., FGF21) in TFAM overexpressing hetero- and homozygotes.

The authors present an excellent technical characterisation of a known mouse model using their proteomic iMPAQT implementation. There are intriguing findings of the stress response, which would merit further study (e.g., SOX6).

A shortcoming is that the metabolomic data could not be clustered into their groups. This is unusual for a phenotypic and in its core "metabolic disease". The use of DNA microarrays for gene expression in contrast to RNAseq seems somewhat old-fashioned.

Unfortunately, the insights into this mouse model are diminished by the fact that they have been reported previously and repeatedly. After this proof-of-principle, venturing in the analysis of a yet uncharacterised mouse model of disease promises major advances in the pathophysiological understanding using their iMPAQT methods. Alternatively, the authors could have extended their analysis to other tissues, which currently are understudied in this respect (e.g., brain).

Overall, this is a nice data report of a known model reported repeatedly and the technical superiority to other proteomics methods would warrant it to be published as a technical report (which has been done before Matsumoto et al., 2017).

Minor comments:

"tended to increase slightly in Tg/Tg"  Please use scientific language.

The authors should include other studies working with TFAM mice (e.g., Bonekamp et al., published in LSA).

Reviewer #1 (Comments to the Authors (Required)):

Igami and colleagues show that mice carrying at least two copies of a transgenic cassette for human TFAM, seemingly one in each chromosome pair, show an extension in lifespan and a decrease in body weight. The authors characterize the muscle and plasma of these mice by transcriptomics, metabolomics, and proteomics to identify a signature that is compatible with the activation of the mitochondrial integrated stress response and an increase in fatty acid and amino acid oxidation. From these data, authors conclude that mitohormesis is induced by high levels of human TFAM expression, inducing metabolic shifts in muscle to explain life span extension. However, there are many technical aspects about this mouse model that are unclear, which impede drawing the conclusions stated by the authors. In addition, other functional analyses key to confirm their conclusions are missing. The major concerns are:

- 1) Both heterozygous and homozygous TFAM mice are defined in the text as TFAM overexpressors, a definition that is confusing and inaccurate. The reason is that Human TFAM is transgenically expressed and not mouse TFAM, so technically it is not an overexpression. To establish whether the phenotype observed is truly induced by differences in TFAM activity, the chromosomal loci where the TFAM transgenic cassettes are located must be defined, to validate that no endogenous genes are disrupted. The fact that the differences are only significant in Tg/Tg mice supports that the disruption of two endogenous alleles by the TFAM cassette, rather than a threshold in TFAM activity, explains the phenotype.

Answer: The gene locus of the vector incorporated into the mouse chromosome is intron 1 region of Irak3 (IRAK-M) between 61986097 and 61990333 on chromosome 10. The published article on this subject can be found in Human Molecular Genetics, 2010, Vol. 19, No. 13 2695-2705 doi:10.1093/hmg/ddq163.

In the mouse skeletal muscle, we confirmed the expression of IRAK-M by Western-blot and found no difference in WT, Tg, and Tg/Tg. Please refer to Response Figure 1.

In this revised manuscript we added the sentence,

“TFAM transgene resided in the intron 1 of the Irak3 gene on chromosome 10(Ylikallio et al., 2010). The protein levels of IRAK-M were not altered in the corresponding mouse models as determined by western-blotting.” (The line 412-415 in page 19)

The above papers have been added as references in the main text. (The line 830-832 in page 37)

- 2) Related to point number 1, it is a possibility that the mitochondrial stress responses observed could be achieved by overexpressing any other human or non-mouse protein (even an inactive form of TFAM), with the phenotype observed is a consequence of proteotoxic stress that results from overexpressing a foreign protein. The lack of a strong phenotype in mice with the human TFAM transgene present only in one chromosome supports this possibility.

Answer: As you say, such a possibility cannot be ruled out, so we added it to the DISCUSSION part of the text,

“Although mitochondrial stress is present in this study, and that an ISR is occurring, there may also be the possibility of proteotoxic stress resulting from overexpression of foreign proteins.” (The line 347-349 in page 16). However, as mentioned in, Ikeda et al. (<https://doi.org/10.1371/journal.pone.0119687>), Matsushima et al. (<https://doi.org/10.1073/pnas.1008924107>) and Bonekamp et al. (<https://doi.org/10.26508/lsa.202101034>) reported that the effect of TFAM overexpression is to repress mt mRNA transcription. We believe that there may be a specific boosting effect of homozygous.

- 3) Human TFAM expressed at very high levels could even behave as a dominant negative form of mouse TFAM. Thus, other mechanisms, rather than over-packing due to supra-physiological TFAM activity as concluded by the authors, can explain the effects observed in TFAM tg/tg mice on mitochondrial gene expression.

Answer: If the suppression of mouse TFAM function as a transcription factor by human TFAM overexpression is occurring, then mtDNA-derived mRNAs would be affected uniformly, which is not actually the case based on the results in Fig. 1f. However, as you say, we cannot completely rule out that possibility at this point, so we added it to the DISCUSSION part of the text,

“It is also undeniable that the excess of human TFAM as a protein may functionally compete with mouse TFAM and impair the original function of mouse TFAM. In such a case, however, mtDNA-derived mRNA would be uniformly affected, but this was not the case (Fig. 1f).” (The line 319-323 in page 15)

- 4) Authors claim that an increase in fatty acid oxidation in muscle can explain the decrease in body weight. However, no measurements of whole-body energy

expenditure, RER and mitochondrial fatty acid oxidation capacity in muscle (or other tissues such as liver or adipose) are shown, which are needed to draw this conclusion of increased fatty acid expenditure.

Answer: Fujii et al. reported the paper(<https://doi.org/10.1016/j.isci.2022.104889>) of Fig.2 that respiratory exchange ratio (RER) was significantly lower in Tg/Tg mice under high-fat diet conditions in the same type of mice experiments.

In this revised manuscript, we added the sentence.

“Fujii et al. reported that respiratory exchange ratios were significantly lower in Tg/Tg mice under high-fat diet conditions(Fujii et al., 2022b), consistent with accelerated fatty acid degradation.” (The lines 373~375, page 17,18)

- 5) Related to point number 4, the authors conclude that a more efficient use of fatty acids for ATP production explains the decrease in FFA content in plasma. However, a more efficient use would have the opposite effect on circulating fatty acid levels: if transgenic mice produce ATP more efficiently from fatty acid oxidation, these mice will utilize less fatty acids to cover the same ATP demand and thus circulating FFA content should not change or even increase. An example is fasting, where fatty acid oxidation is increased in muscle and liver, but circulating levels are not decreased. In this regard, TFAM overexpression in adipose tissue could impair lipolysis and white adipose tissue function to decrease circulating fatty acid levels.

Answer: We reconsidered why the plasma FFA was reduced as you stated, the Tg/Tg mice were basically quite lean, and it was obvious that they had low fat mass throughout their bodies. We simply assumed that this phenotype was the reason for the low plasma FFA. We believe that the relationship between ATP production and free FFA in blood is tenuous, so we have removed the relevant statements from the RESULT and DISCUSSION.

In this revised manuscript, we added the sentence.

“Dissection revealed less fat and they also had low levels of free fatty acids in their blood. (Fig. S1)” (The lines 148~149, page 8)

We added the Supplementary Figure 1. The data for FFA in plasma in Fig.4e are transferred here. (The lines 985~993, page 45)

- 6) Do the authors think that the elevation of fatty acid oxidation is a consequence of the increase in FGF21 or just because higher TFAM content in mitochondria induces a direct effect on them to elevate their fatty acid oxidation capacity? Given that the TFAM tg/tg mouse line is a whole-body transgenic mouse, how much of the lifespan

extension stems from human TFAM actions in the muscle, versus human TFAM actions in other tissues? These issues should be discussed in the manuscript, as well as toning down the conclusions on the contributions of TFAM actions in muscle to the phenotype.

Answer: We believe it is the former you mentioned. That is, TFAM overexpression → constant mitochondrial stress (increased FGF21, other stress signals) → mitohormesis results in improved fatty acid degradation as a phenotype. The extent to which the phenotype observed in this study contributes to the lifespan extension effect will be elucidated in future studies, but the fact that human TFAM expression is induced by the β -actin promoter suggests that there must be at least some muscle-derived causal relationship.

In this revised manuscript, we added the sentence.

“Since the TFAM Tg/Tg mouse is a whole-body transgenic mouse, the extent to which the skeletal muscle phenotype observed in this study contributes to the lifespan-extending effect should be clarified in future studies.” (The lines 403~405, page 19)

- 7) Gastrocnemius is not a fast twitch muscle, it has both oxidative and glycolytic fibers.

Answer: I have changed the text to the following sentence as you stated.

“The mouse gastrocnemius muscle has both oxidative and glycolytic fibers muscle that uses glycolysis to provide energy,” (The lines 291, page 14)

- 8) Authors suggest that TFAM tg/tg might show muscle atrophy, given the signature of increased amino acid oxidation and decreased protein translation characteristic of ISR activation. How do authors reconcile these findings with increased lifespan? Would this increase in lifespan be meaningful for a wild mouse if the transgenic mouse cannot run fast and escape its predators?

Answer: Since there is atrophy of muscle tissue at Tg/Tg observed in this study, as you say, the athletic ability to generate speed may be reduced and survival in the wild may be reduced. We thought that having skeletal muscle that is more prone to fat and BCAA breakdown might be akin to forced caloric restriction, and that this might influence lifespan progression.

The result is Tg/Tg mice with small body size, which is interesting because it is like snell dwaf mice with small body size and life span progression. (<https://doi.org/10.1111/ace1.13030> In this report, it is reported that the expression of

TFAM is increased in snell cells as a stress response.)

- 9) Authors show a decrease in the mRNA content of different OXPHOS components, while proteomics data reveal an accumulation of some OXPHOS proteins that showed a decrease in their mRNA levels. Authors conclude that this discrepancy between mRNA and protein can reflect mitohormesis and the ISR activation. However, such an increase in protein content in samples in which the ISR is activated, which decreases protein translation, will most likely be explained by a selective blockage of the degradation of some mitochondrial proteins. In addition, macroautophagy can also increase amino acid and fatty acid catabolism, with changes in mitochondrial shape being demonstrated to spare mitochondria from degradation when autophagy is activated. Thus, authors should either block the ISR with ISRIB or block its metabolic effector FGF21, GDF15, to test the dependency of the phenotype on the ISR, or measure autophagy and mitochondrial shape, to determine whether those could explain the phenotype.

Answer: For the OXPHOS complex genes encoded in mtDNA (present in complexes 1, 3, and 4), those with lowered mRNA were similarly downregulated in the protein measurement results by iMPAQT.

Complex 2 is a gene (nuclear code) that is not present in mitochondrial DNA, so the results of protein expression by iMPAQT are not contradictory to begin with. We also believe that this is not a discrepancy between gene and protein expression, since the subunits of complex 5 (ATP-ase) measured by iMPAQT were derived from the nucleus.

We believe that at least a decrease in the mtDNA-encoded 1,3,4 respiratory chain complex is certain and is related to TFAM overexpression, an mtDNA transcription factor.

The relationship between phenotype and ISR is, as you say, a very important discussion, and I would like to address it as the main theme of my next research project.

Reviewer #2 (Comments to the Authors (Required)):

The paper addresses a unique role of a mitochondrial transcription factor TFAM in mitohormesis and resulting healthier phenotype. The authors performed the gene expression analysis which revealed mitochondrial stress in TFAM-overexpressing skeletal muscle leading to a beneficial adaptive response and life extension in mice. Proteomics and metabolomics were employed to demonstrate that overexpressing the TFAM gene in mice induced a metabolic shift towards enhanced degradation of fatty acids (FA) and branched-chain amino acids (BCAA) and decreased glycolysis. The presented data justified the conclusions of the paper but the following revisions should be considered.

1. While the iMPAQT method provides quantitative protein analysis, the LC-MS and GC-MS analyses of metabolites are qualitative. The targeted, quantitative LC-MS/MS or GC-MS analyses should be used to confirm that major metabolic intermediates of TCA cycle and FA and BCAA metabolism pathways are affected by the TFAM overexpression. In particular, branched-chain keto acids should be also analyzed in addition to BCAA to prove the increased degradation of BCAA in Tg/Tg mice.

Answer: We have results detected by both LC-MS and GC-MS for metabolites in the glycolysis, TCA cycle, and BCAA degradation pathways. Please see Response Fig. 2. We have confirmed that these metabolites show a common behavior even when using different mass spectrometry methods. Although this is not an absolute quantitative result, we believe that the results are very reproducible. The GC-MS results for branched-chain keto acids (oxoisocaproate, 3-Methyl-2-oxopentanoate, Oxoisovalerate) have been newly added, and are shown in Fig. 5 a and c.

2. The paper states enhanced fatty acid degradation pathway in skeletal muscle of TFAM-overexpressing mice. How do the authors explain the fact that while the increase of acetyl CoA was observed in Tg/Tg mice, acetyl carnitine showed no change?

Answer: We do not believe that this is a situation where fatty acid degradation is constantly active, and that the rise in acetyl CoA tends to be more dependent on BCAA degradation. As it relates to the answer to your next question, we are considering the possibility of a situation where the total fat content itself is already low and metabolites derived from fatty acid degradation are low as a Tg/Tg phenotype.

3. The mass spec analysis showed decreased levels of FFA in Tg/Tg plasma and

increased levels of many hydroxylated acylcarnitine species in Tg/Tg skeletal muscle, however the more abundant acylcarnitines such as palmitoyl carnitine and oleoyl carnitine that come from fatty acid oxidation showed no change. How does this agree with the statement that the fatty oxidation pathway is enhanced by the TFAM overexpression?

Answer: Considering the suggestions you have pointed out, we have changed the interpretation of the results shown in Fig. 4e, which showed a decrease in free fatty acids in the blood at Tg/Tg. We considered that this result simply reflected a lower total fat content in Tg/Tg. In fact, we also observed less fat when the Tg/Tg mice were dissected, and we added this result to the blood free fatty acids results as a new Supplement Fig. 1. We interpret the genetic and protein results as indicating that fat is easily degraded in the muscle, but since fat is already burned off, it was difficult to see that part of the results from the metabolomics results.

In this revised manuscript, we added the sentence.

“Dissection revealed less fat and they also had low levels of free fatty acids in their blood. (Fig. S1)” (The lines 148~149, page 8)

We added the Supplementary Figure 1. The data for FFA in plasma in Fig.4e are transferred here. (The lines 985~993, page 45)

4. The authors suggest that Tg/Tg mice may have a permanent shortage of energy. To test this hypothesis, ATP and AMP levels should be analyzed in the skeletal muscle.

Answer: AMP, ADP, and ATP measurements have been added to Fig. 6g in the revised paper. Please check them.

AMP and ADP were increased in Tg/Tg mice, but despite this, there was a decreasing trend in ATP, which we believe is consistent with our previous assertion.

In this revised manuscript, we added the sentence.

“In addition, a decreasing trend was observed for ATP, even though AMP and ADP were increasing at Tg/Tg mice (Fig. 6g).” (The lines 298~300, page 14)

We added the words to the following sentence.

“(e) Overview of the proteins and metabolites detected in the creatine and ATP pathway.” (The lines 973~974, page 43)

5. The authors state that BCAAs are rarely metabolized in the liver and are instead metabolized in tissues, primarily skeletal muscle. This statement needs to be double checked as there are many published reports showing BCAA metabolism in liver.

Answer: In this revised manuscript, we have changed the text as follows.

“BCAAs are often metabolized in tissues such as skeletal muscle.” (The lines

376~377, page 18)

Reviewer #3 (Comments to the Authors (Required)):

Igami et al. present a comprehensive omics analysis of TFAM overexpressing mice strains (Tg and Tg/Tg) and identify respective metabolic adaptations. Using their iMPAQT pipeline for absolute protein quantification published elsewhere they report that TFAM overexpression results in mitohormesis, increased FA and AA degradation and decreased glycolysis in skeletal muscle predominantly. They furthermore integrate and crosscorrelate metabolomic data and use DNA microarray gene expression and report typical mitochondrial stress response markers (e.g., FGF21) in TFMA overexpressing hetero- and homozygotes.

The authors present an excellent technical characterisation of a known mouse model using their proteomic iMPAQT implementation. There are intriguing findings of the stress response, which would merit further study (e.g., SOX6).

A shortcoming is that the metabolomic data could not clustered into their groups. This is unusual for a phenotypic and in its core "metabolic disease". The use of DNA microarrays for gene expression in contrast to RNAseq seems somewhat old-fashioned. (Our comment) Metabolites have many immediate factors, and as you say, it was disappointing that we could not discriminate Tg/Tg mice as a result of the overall metabolomics. DNA microarrays are, as you say, a bit technologically archaic, but the database used in the GSEA analysis was the latest one uploaded in 2023.

Unfortunately, the insights into this mouse model are diminished by the fact that they have been reported previously and repeatedly. After this proof-of-principle, venturing in the analysis of a yet uncharacterised mouse models of disease promises major advances in the pathophysiological understanding using their iMPAQT methods. Alternatively, the authors could have extended their analysis to other tissues, which currently are understudied in this respect (e.g., brain).

(Our comment) As you mentioned, after this proof-of-principle demonstration, we are working on various unique materials as the next application of the iMPAQT method. Neuronal cells are one such application, and we hope that you will look forward to it.

Overall, this is a nice data report of a known model reported repeatedly and the technical superiority to other proteomics method would warrant it to published as technical report (which has been done before Matsumoto et al., 2017).

(Our comment) In particular, we believe that the raw proteomics data should be published for future research, and we have repositied it on the PanoramaWeb below.

In this revised manuscript, we added the sentence.

“The MS proteomics data have been deposited to the PanoramaWeb (<https://panoramaweb.org>) with the data set Permanent Link: <https://panoramaweb.org/GqfOgy.url>.” (The lines 877~879, page 39)

Minor comments:

"tended to increase slightly in Tg/Tg"  Please use scientific language.

Answer: In this revised manuscript, we have changed the text as follows.

“In the TCA cycle, both protein and metabolites were slightly increased in Tg/Tg mice compared with WT” (The lines 285, page 14)

The authors should include other studies working with TFAM mice (e.g., Bonekamp et al., published in LSA).

Answer: In this revised manuscript we added the sentence,

“Recent reports have also shown that very high TFAM expression in mice suppresses mitochondrial DNA transcription (Bonekamp et al., 2021).” (The line 315-316 in page 15)

The above papers have been added as references in the main text. (The line 612-615 in page 29)

Response Fig. 1 for Reviewer 1

Western blotting targeting IRAK-M from mouse skeletal muscle

IRAK-M : Anti-IRAK-M produced in rabbit, affinity isolated antibody
(I5157, MilliporeSigma)

β-actin : Monoclonal Anti-b-Actin Clone AC-15(A5441, MilliporeSigma)

Response Fig. 2 for Reviewer 2

Glycolysis

TCA cycle

Valine, leucine and isoleucine degradation

April 9, 2024

RE: Life Science Alliance Manuscript #LSA-2023-02498-TR

Dr. Ko Igami
LSI Medience corp.
2-3, Shibaura 1-chome
Minato-ku, Tokyo 105-0023
Japan

Dear Dr. Igami,

Thank you for submitting your revised manuscript entitled "iMPAQT reveals that adequate mitohormesis from TFAM overexpression leads to life extension in mice". We would be happy to publish your paper in Life Science Alliance pending final revisions necessary to meet our formatting guidelines.

- please address Reviewer 1's remaining comments
- please be sure that the authorship listing and order is correct
- please add the Twitter handle of your host institute/organization as well as your own or/and one of the authors in our system
- please use the [10 author names et al.] format in your references (i.e., limit the author names to the first 10)
- the Reference section should be moved to after the Data Availability statement
- please add callouts for Figure S1A and B to your main manuscript text

A. FINAL FILES:

B. MANUSCRIPT ORGANIZATION AND FORMATTING:

**Submission of a paper that does not conform to Life Science Alliance guidelines will delay the acceptance of your

manuscript.**

The license to publish form must be signed before your manuscript can be sent to production. A link to the electronic license to publish form will be available to the corresponding author only. Please take a moment to check your funder requirements.

Sincerely,

Reviewer #1 (Comments to the Authors (Required)):

The authors responded to most of the concerns successfully, by providing key data on the locus of insertion and toning down the conclusions that were not supported by data. The minor concerns that remain are as follows:

- 1)The Western blot data measuring IRAK content needs to be included in the manuscript, in one of the main figures. It is an essential control to prove that the phenotype observed is by an additional increase in human TFAM content.
- 2) In the text, there are still instances where it is concluded that TFAM overexpression → constant mitochondrial stress (increased FGF21, other stress signals) → mitohormesis results in improved fatty acid degradation as a phenotype. However, the accumulation of OXPHOS proteins strongly supports that changes in autophagy could be driving the effects. Authors should acknowledge better in the text that ISR activation and mitohormesis might not be the drivers of the phenotype.

Reviewer #2 (Comments to the Authors (Required)):

All my comments have been adequately addressed in the revised version of the manuscript.

Reviewer #1 (Comments to the Authors (Required)):

The authors responded to most of the concerns successfully, by providing key data on the locus of insertion and toning down the conclusions that were not supported by data. The minor concerns that remain are as follows:

1)The Western blot data measuring IRAK content needs to be included in the manuscript, in one of the main figures. It is an essential control to prove that the phenotype observed is by an additional increase in human TFAM content.

Answer: In this revision, Western blot data of IRAK-M is added to Fig. 1e.

In addition, the following note has been added to the manuscript.

“On the other hand, the TFAM transgene was located in intron 1 of the Irak3 (IRAK-M) gene on chromosome 10(Ylikallio et al., 2010). Therefore, we confirmed the protein level of IRAK-M expression in gastrocnemius muscle by Western blotting and found no changes in both Tg and Tg/Tg (Fig. 1e).” (The line 152-156 in page 8)

“Immunoblotting” is added to Methods chapter. (The line 427-439 in page 20)

2) In the text, there are still instances where it is concluded that TFAM overexpression → constant mitochondrial stress (increased FGF21, other stress signals) → mitohormesis results in improved fatty acid degradation as a phenotype. However, the accumulation of OXPHOS proteins strongly supports that changes in autophagy could be driving the effects. Authors should acknowledge better in the text that ISR activation and mitohormesis might not be the drivers of the phenotype.

Answer: Since this is a possibility, as you mentioned, I have added the following text to the DISCUSSION section.

“On the other hand, there is also the possibility of other biological mechanisms that can lead to this phenotype (e.g., autophagy changes promote the effect because of the accumulation of OXPHOS proteins), which needs to be verified.” (The line 409-412 in page 19)

One additional reference that mentions the relationship between mitochondrial failure and ISR has been added. (The line 333-334 in page 16)

April 16, 2024

RE: Life Science Alliance Manuscript #LSA-2023-02498-TRR

Dr. Ko Igami
LSI Medience corp.
2-3, Shibaura 1-chome
Minato-ku, Tokyo 105-0023
Japan

Dear Dr. Igami,

Thank you for submitting your Resource entitled "iMPAQT reveals that adequate mitohormesis from TFAM overexpression leads to life extension in mice". It is a pleasure to let you know that your manuscript is now accepted for publication in Life Science Alliance. Congratulations on this interesting work.

DISTRIBUTION OF MATERIALS:

Again, congratulations on a very nice paper. I hope you found the review process to be constructive and are pleased with how the manuscript was handled editorially. We look forward to future exciting submissions from your lab.

Sincerely,
